# Watch the Unobserved: A Simple Approach to Parallelizing Monte Carlo Tree Search

**Anji Liu**[†]**, Jianshu Chen**[⋆]**, Mingze Yu**[†]**, Yu Zhai**[†]**, Xuewen Zhou**[†] **& Ji Liu**[†]
[†]Seattle AI Lab, Kwai Inc., Bellevue, WA 98004, USA
`{liuanji03,yumingze,zhaiyu,zhouxuewen,jiliu}@kuaishou.com`
[⋆]Tencent AI Lab, Bellevue, WA 98004, USA
`jianshuchen@tencent.com`

## Abstract

Monte Carlo Tree Search (MCTS) algorithms have achieved great success on many challenging benchmarks (e.g., Computer Go). However, they generally require a large number of rollouts, making their applications costly. Furthermore, it is also extremely challenging to parallelize MCTS due to its inherent sequential nature: each rollout heavily relies on the statistics (e.g., node visitation counts) estimated from previous simulations to achieve an effective exploration-exploitation tradeoff. In spite of these difficulties, we develop an algorithm, *WU-UCT*[1], to effectively parallelize MCTS, which achieves linear speedup and exhibits only limited performance loss with an increasing number of workers. The key idea in WU-UCT is a set of statistics that we introduce to track the number of on-going yet incomplete simulation queries (named as *unobserved samples*). These statistics are used to modify the UCT tree policy in the selection steps in a principled manner to retain effective exploration-exploitation tradeoff when we parallelize the most time-consuming expansion and simulation steps. Experiments on a proprietary benchmark and the Atari Game benchmark demonstrate the linear speedup and the superior performance of WU-UCT comparing to existing techniques.

## 1 Introduction

Recently, Monte Carlo Tree Search (MCTS) algorithms such as UCT (Kocsis et al., 2006) have achieved great success in solving many challenging artificial intelligence (AI) benchmarks, including video games (Guo et al., 2016) and Go (Silver et al., 2016). However, they rely on a large number (e.g. millions) of interactions with the environment emulator to construct search trees for decision-making, which leads to high time complexity (Browne et al., 2012). For this reason, there has been an increasing demand for parallelizing MCTS over multiple workers. However, parallelizing MCTS without degrading its performance is difficult (Segal, 2010; Mirsoleimani et al., 2018a; Chaslot et al., 2008), mainly due to the fact that each MCTS iteration requires information from all previous iterations to provide effective exploration-exploitation tradeoff. Specifically, parallelizing MCTS would inevitably obscure these crucial information, and we will show in Section 2.2 that this loss of information potentially results in a significant performance drop. The key question is therefore how to acquire and utilize more available information to mitigate the information loss caused by parallelization and help the algorithm to achieve better exploration-exploitation tradeoff.

To this end, we propose WU-UCT (**W**atch the **U**novserved in **UCT**), a novel parallel MCTS algorithm that attains linear speedup with only limited performance loss. This is achieved by a conceptual innovation (Section 3.1) as well as an efficient real system implementation (Section 3.2). Specifically, the key idea in WU-UCT to overcome the aforementioned challenge is a set of statistics that tracks the number of on-going yet incomplete simulation queries (named as *unobserved samples*). We combine these newly devised statistics with the original statistics of *observed samples* to modify UCT's policy in the selection steps in a principled manner, which, as we shall show in Section 4, effectively retains exploration-exploitation tradeoff during parallelization. Our proposed

---

[1]Code is available at `https://github.com/liuanji/WU-UCT`.

approach has been successfully deployed in a production system for *efficiently* and *accurately* estimating the rate at which users pass levels (termed *user pass-rate*) in a mobile game "Joy City", with the purpose of reducing their design cycles. On this benchmark, we show that WU-UCT achieves near-optimal linear speedup and superior performance in predicting user pass-rate (Section 5.1). We further evaluate WU-UCT on the Atari Game benchmark and compare it to state-of-the-art parallel MCTS algorithms (Section 5.2), which also demonstrate our superior speedup and performance.

## 2 ON THE DIFFICULTIES OF PARALLELIZING MCTS

We first introduce the MCTS and the UCT algorithms, along with their difficulties in parallelization.

### 2.1 MONTE CARLO TREE SEARCH AND UPPER CONFIDENCE BOUND FOR TREES (UCT)

We consider the Markov Decision Process (MDP) $\langle \mathcal{S}, \mathcal{A}, R, P, \gamma \rangle$, where an agent interacts with the environment in order to maximize a long-term cumulative reward. Specifically, an agent at state $s_t \in \mathcal{S}$ takes an action $a_t \in \mathcal{A}$ according to a *policy* $\pi$, so that the MDP transits to the next state $s_{t+1} \sim P(s_{t+1}|s_t, a_t)$ and emits a reward $R(s_t, a_t)$.[2] The objective of the agent is to learn an optimal policy $\pi^*$ such that the long-term cumulative reward is maximized:

$$\max_{\pi} \mathbb{E}_{a_t \sim \pi, s_{t+1} \sim P} \Big[ \sum_{t=0}^{\infty} \gamma^t R(s_t, a_t) \,|\, s_0 = s \Big], \tag{1}$$

where $s \in \mathcal{S}$ denotes the initial state and $\gamma \in (0, 1]$ is the discount factor.[3] Many reinforcement learning (RL) algorithms have been developed to solve the above problem (Sutton & Barto, 2018), including model-free algorithms (Mnih et al., 2013; 2016; Williams, 1992; Konda & Tsitsiklis, 2000; Schulman et al., 2015; 2017) and model-based algorithms (Nagabandi et al., 2018; Weber et al., 2017; Bertsekas, 2005; Deisenroth & Rasmussen, 2011). Monte Carlo Tree Search (MCTS) is a model-based RL algorithm that *plans* the best action at each time step (Browne et al., 2012). Specifically, it uses the MDP model (or its sampler) to identify the best action at each time step by constructing a search tree (Figure 1(a)), where each node $s$ represents a visited state, each edge from $s$ denotes an action $a_s$ that can be taken at that state, and the landing node $s'$ denotes the state it transits to after taking $a_s$. As shown in Figure 1(a), MCTS repeatedly performs four *sequential* steps: selection, expansion, simulation and backpropagation. The selection step traverses over the existing search tree until the leaf node (or other termination conditions are satisfied) by choosing actions (edges) $a_s$ at each node $s$ according to a tree policy. One widely used node-selection policy is the one used in the Upper Confidence bound for Trees (UCT) (Kocsis et al., 2006):

$$a_s = \arg\max_{s' \in \mathcal{C}(s)} \left\{ V_{s'} + \beta \sqrt{\frac{2 \log N_s}{N_{s'}}} \right\}, \tag{2}$$

where $\mathcal{C}(s)$ denotes the set of all child nodes for $s$; the first term $V_{s'}$ is an estimate for the long-term cumulative reward that can be received when starting from the state represented by node $s'$, and the second term represents the uncertainty (size of the confidence interval) of that estimate. The confidence interval is calculated based on the Upper Confidence Bound (UCB) (Auer et al., 2002; Auer, 2002) using $N_s$ and $N_{s'}$, which denote the number of times that the nodes $s$ and $s'$ have been visited, respectively. Therefore, the key idea of the UCT policy (2) is to select the best action according to an optimistic estimation (i.e., the upper confidence bound) of the expected return, which strikes a balance between the exploitation (first term) and the exploration (second term) with $\beta$ controlling their tradeoff. Once the selection process reaches a leaf node of the search tree (or other termination conditions are met), we will expand the node according to a prior policy by adding a new child node. Then, in the simulation step, we estimate its value function (cumulative reward) $\hat{V}_s$ by running the environment simulator with a default (simulation) policy. Finally, during *backpropagation*, we update the statistics $V_s$ and $N_s$ from the leaf node $s_T$ to the root node $s_0$ of the selected path by recursively performing the following update (i.e., from $t = T - 1$ to $t = 0$):

$$N_{s_t} \leftarrow N_{s_t} + 1, \quad \hat{V}_{s_t} \leftarrow R(s_t, a_t) + \gamma \hat{V}_{s_{t+1}}, \quad V_{s_t} \leftarrow \big((N_{s_t} - 1)V_{s_t} + \hat{V}_{s_t}\big)/N_{s_t}, \tag{3}$$

---

[2]In the context of MCTS, the action space $\mathcal{A}$ is assumed to be finite and the transition $P$ is assumed to be deterministic, i.e., the next state $s_{t+1}$ is determined by the current state $s_t$ and action $a_t$.

[3]We assume certain regularity conditions hold so that the cumulative reward $\sum_{t=0}^{\infty} \gamma^t R(s_t, a_t)$ is always bounded (Sutton & Barto, 2018).

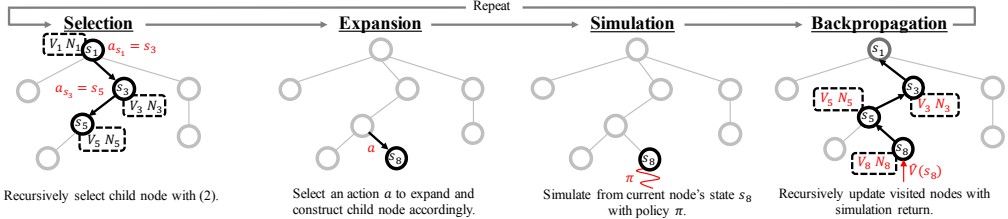

(a) Each (non-parallel) MCTS rollout consists of four *sequential* steps: selection, expansion, simulation and backpropagation, where the expansion and the simulation steps are generally most time-consuming.

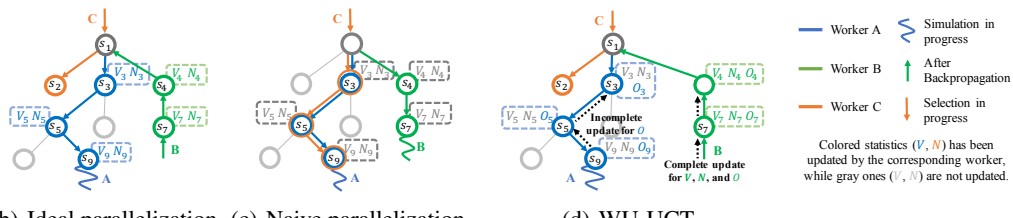

(b) Ideal parallelization    (c) Naive parallelization        (d) WU-UCT

Figure 1: MCTS and its parallelization. (a) An overview of MCTS. (b) The ideal parallelization: the most up-to-date statistics $\{V_s, N_s\}$ (in chromatic color) are assumed to be available to all workers as soon as a simulation begins (unrealistic in practice). (c) The key challenge in parallelizing MCTS: the workers can only access outdated $\{V_s, N_s\}$ (in gray-color), leading problems like *collapse of exploration*. (d) WU-UCT tracks the number of incomplete simulation queries, which is denoted as $O_s$, and modifies the UCT policy in a principled manner to retain effective exploration-exploitation tradeoff. It achieves comparable speedup and performance as the ideal parallelization.

where $\hat{V}_{s_T}$ is the simulation return of $s_T$; $a_t$ denotes the action selected following (2) at state $s_t$.

## 2.2 THE INTRINSIC DIFFICULTIES OF PARALLELIZING MCTS

The above discussion implies that the MCTS algorithm is intrinsically sequential: each selection step in a new rollout requires the previous rollouts to complete in order to deliver the updated statistics, $V_s$ and $N_s$, for the UCT tree policy (2). Although this requirement of up-to-date statistics is not mandatory for implementation, it is in practice intensively required to achieve effective exploration-exploitation tradeoff (Auer et al., 2002). Specifically, up-to-date statistics best help the UCT tree policy to identify and prune non-rewarding branches as well as extensively visiting rewarding paths for additional planning depth. Likewise, to achieve the best possible performance, when multiple workers are used, it is also important to ensure that each worker uses the most recent statistics (the colored $V_s$ and $N_s$ in Figure 1(b)) in its own selection step. However, this is impossible in parallelizing MCTS based on the following observations. First, the expansion step and the simulation step are generally more time-consuming compared to the other two steps, because they involve a large number of interactions with the environment (or its simulator). Therefore, as exemplified by Figure 1(c), when a worker C initiates a new selection step, the other workers A and B are most likely still in their simulation or expansion steps. This prevents them from updating the (global) statistics for other workers like C, which happens at their respective backpropagation steps. Using outdated statistics (the gray-colored $V_s$ and $N_s$) at different workers could lead to a significant performance loss given a fixed target speedup, due to behaviors like *collapes of exploration* or *exploitation failure*, which we shall discuss thoroughly in Section 4. To give an example, Figure 1(c) illustrates the collapse of exploration, where worker C traverses over the same path as the worker A in its selection step due to the determinism of (2). Specifically, if the statistics are unchanged between the moments that worker A and C begin their own selection steps, they will choose the same node according to (2), which greatly reduces the diversity of exploration. Therefore, the key question that we want to address in parallelizing MCTS is how to track the correct statistics and modify the UCT policy in a *principled* manner, with the hope of retaining effective exploration-exploitation tradeoff at different workers.

## 3   WU-UCT

In this section, we first develop the conceptual idea of our WU-UCT algorithm (Section 3.1), and then we present a real system implementation using a master-worker architecture (Section 3.2).

### 3.1   WATCH THE UNOBSERVED SAMPLES IN UCT TREE POLICY

As we pointed out earlier, the key question we want to address in parallelizing MCTS is how to deliver the most up-to-date statistics $\{V_s, N_s\}$ to each worker so that they can achieve effective exploration-exploitation tradeoff in its selection step. This is assumed to be the case in the ideal parallelization in Figure 1(b). Algorithmically, it is equivalent to the sequential MCTS except that the rollouts are performed in parallel by different workers. Unfortunately, in practice, the statistics $\{V_s, N_s\}$ available to each worker are generally outdated because of the slow and incomplete simulation and expansion steps at the other workers. Specifically, since the estimated value $\hat{V}_s$ is unobservable before simulations complete and workers should not wait for the updated statistics to proceed, the (partial) loss of statistics $\{V_s, N_s\}$ is unavoidable. Now the question becomes: is there an alternative way to addressing the issue? The answer is in the affirmative and is explained below.

Aiming at bridging the gap between naive parallelization and the ideal case, we closely examine their difference in terms of the availability of statistics. As illustrated by the colors of the statistics, their only difference in $\{V_s, N_s\}$ is caused by the on-going simulation process. As suggested by (3), although $V_s$ can only be updated after a simulation step is completed, the newest $N_s$ information can actually be available as early as a worker initiates a new rollout. This is the key insight that we leverage to enable effective parallelization in our WU-UCT algorithm. Motivated by this, we introduce another quantity, $O_s$, to count *the number of rollouts that have been initiated but not yet completed*, which we name as *unobserved samples*. That is, our new statistics, $O_s$, watch the number of unobserved samples, and are then used to correct the UCT tree policy (2) into the following form:

$$a_s = \underset{s' \in \mathcal{C}(s)}{\arg\max} \left\{ V_{s'} + \beta \sqrt{\frac{2\log(N_s + O_s)}{N_{s'} + O_{s'}}} \right\}. \tag{4}$$

The intuition of the above modified node-selection policy is that when there are $O_s$ workers simulating (querying) node $s$, the confidence interval at node $s$ will eventually be shrunk after they complete. Therefore, adding $O_s$ and $O_{s'}$ to the exploration term considers such a fact beforehand and let other workers be aware of it. Despite its simple form, (4) provides a principled way to retain effective exploration-exploitation tradeoff under parallel settings; it corrects the confidence bound towards better exploration-exploitation tradeoff. As the confidence level is instantly updated (i.e., at the beginning of simulation), more recent workers are guaranteed to observe additional statistics, which prevent them from extensively querying the same node as well as find better nodes for them to query. For example, when multiple children are in demand for exploration, (4) allows them to be explored evenly. In contrast, when a node has been sufficiently visited (i.e., large $N_s$ and $N_{s'}$), adding $O_s$ and $O_{s'}$ from the unobserved samples have little effect on (4) because the confidence interval is sufficiently shrunk around $V_{s'}$, allowing extensively exploitation of the best-valued child.

### 3.2   SYSTEM IMPLEMENTATION USING MASTER-WORKER ARCHITECTURES

We now proceed to explain the system implementation of WU-UCT, where the overall architecture is shown in Figure 2(a) (see Appendix A for the details). Specifically, we use a master-worker architecture to implement the WU-UCT algorithm with the following considerations. First, since the expansion and the simulation steps are much more time-consuming compared to the selection and the backpropagation steps, they should be intensively parallelized. In fact, they are relatively easy to parallelize (e.g., different simulations could be performed independently). Second, as we discussed earlier, different workers need to access the most up-to-date statistics $\{V_s, N_s, O_s\}$ in order to achieve successful exploration-exploitation tradeoff. To this end, a centralized architecture for the selection and backpropagation step is more preferable as it allows adding strict restrictions to the retrieval and update of the statistics, making them up-to-date. Specifically, we use a centralized master process to maintain a *global* set of statistics (in addition to other data such as game states), and let it be in charge of the backpropagation step (i.e., updating the global statistics) and the selection step (i.e., exploiting the global statistics). As shown in Figure 2(a), the master process

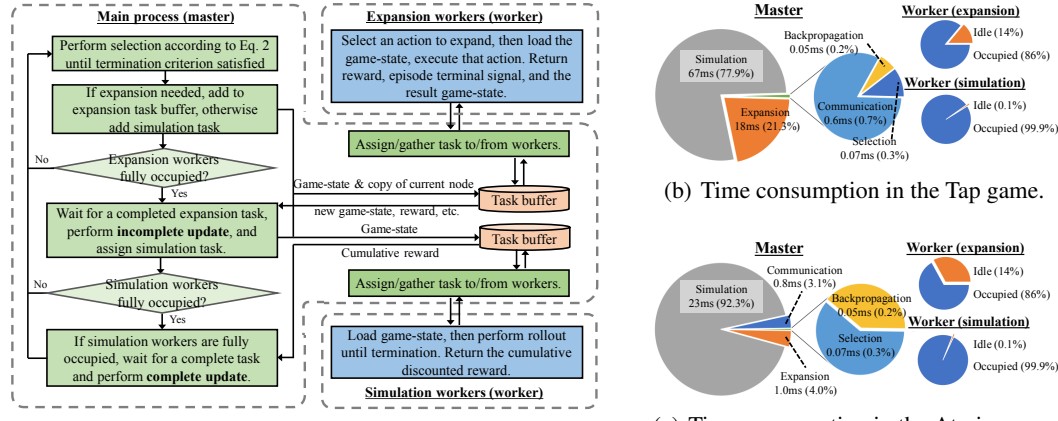

(a) The system architecture that implements WU-UCT.

(b) Time consumption in the Tap game.

(c) Time consumption in the Atari games.

Figure 2: Diagram of WU-UCT's system architecture and its time consumption. (a) the Green blocks and the task buffers are operated at the master, while the blue blocks are executed by the workers. (b-c) the breakdown of the time consumption on two game benchmarks (Section 5).

repeatedly performs rollouts until a predefined number of simulations is reached. During each roll-out, it selects nodes to query, assign expansion and simulation tasks to different workers, and collect the returned results to update the global statistics. In particular, we use the following *incomplete update* and *complete update* (shown in Figure 2(a)) to track $N_s$ and $O_s$ along the traversed path (see Figure 1(d)):

$$[\text{incomplete update}] \quad O_s \leftarrow O_s + 1, \tag{5}$$

$$[\text{complete update}] \quad O_s \leftarrow O_s - 1; \ N_s \leftarrow N_s + 1, \tag{6}$$

where *incomplete update* is performed before the simulation task starts, allowing the updated statistics to be instantly available globally; *complete update* is done after the simulation return is available, resembling the backpropagation step in the sequential algorithm. In addition, $V_s$ is also updated in the complete update step using (3). Such a clear division of labor between the master and the workers provides sequential selection and backpropagation steps when we parallelize the costly expansion and simulation steps. It ensures up-to-date statistics for all workers by the centralized master process and achieves linear speedup without much performance degradation (see Section 5 for the experimental results).

To justify the above rationale of our system design, we perform a set of running time analysis for our developed WU-UCT system and report the results in Figure 2(b)–(c). We show the time-consumption for different parts at the master and at the workers. First, we focus exclusively on the workers. With a close-to-100% occupancy rate for the simulation workers, the simulation step is fully parallelized. Although the expansion workers are not fully utilized, the expansion step is maximumly parallelized since the number of required simulation and expansion tasks is identical. This suggests the existence of an optimal (task-dependent) ratio between the number of expansion workers and the number of simulation workers that fully parallelize both steps with the least resources (e.g. memory). Returning to the master process, on both benchmarks, we see a clear dominance of the time spent on the simulation and the expansion steps even they are both parallelized by 16 workers. This supports our design to parallelize only the simulation and expansion steps. We finally focus on the communication overhead caused by parallelization. Although more time-consuming compared to simulation and backpropagation, the communication overhead is negligible compared to the time used by the expansion and the simulation steps. Other details in our system, such as the centralized game-state storage, are further discussed in Appendix A.

## 4 THE BENEFITS OF WATCHING UNOBSERVED SAMPLES

In this section, we discuss the benefits of watching unobserved samples in WU-UCT, and compare it with several popular parallel MCTS algorithms (Figure 3), including Leaf Parallelization (LeafP),

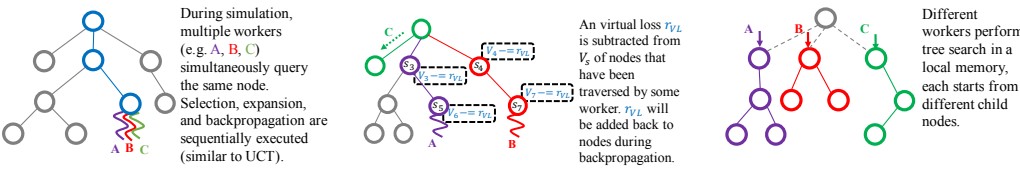

(a) Leaf parallelization (LeafP)     (b) Tree parallelization (TreeP)     (c) Root parallelization (RootP)

Figure 3: Three popular parallel MCTS algorithms. LeafP parallelizes the simulation steps, TreeP uses virtual loss to encourage exploration, and RootP parallelizes the subtrees of the root node.

Tree Parallelization (TreeP) with virtual loss, and Root Parallelization (RootP).[4] LeafP parallelizes the leaf simulation, which leads to an effective hex game solver (Wang et al., 2018). TreeP with virtual loss has recently achieved great success in challenging real-world tasks such as Go (Silver et al., 2016). And RootP parallelizes the subtrees of the root node at different workers and aggregates the statistics of the subtrees after all the workers complete their simulations (Soejima et al., 2010).

We argue that, by introducing the additional statistics $O_s$, WU-UCT achieves a better exploration-exploitation tradeoff than the above methods. First, LeafP and TreeP represent two extremes in such a tradeoff. LeafP lacks diversity in exploration as all its workers are assigned to simulating the same node, leading to performance drop caused by *collapse of exploration* in much the same way as the naive parallelization (see Figure 1(c)). In contrast, although the virtual loss used in TreeP could encourage exploration diversity, this hard additive penalty could cause *exploitatin failure*: workers will be less likely to co-simulating the same node even when they are certain that it is optimal (Mirsoleimani et al., 2017). RootP tries to avoid these issues by letting workers perform an independent tree search. However, this reduces the equivalent number of rollouts at each worker, decreasing the accuracy of the UCT policy (2). Different from the above three approaches, WU-UCT achieves a much better exploration-exploitation tradeoff in the following manner. It encourages exploration by using $O_s$ to "penalize" the nodes that have many in-progress simulations. Meanwhile, it allows multiple workers to exploit the most rewarding node since this "penalty" vanishes when $N_s$ becomes large (see (4)).

## 5 EXPERIMENTS

This section evaluates the proposed WU-UCT algorithm on a production system to predict the user pass-rate of a mobile game (Section 5.1) as well as on the public Atari Game benchmark (Section 5.2), aiming at demonstrating the superior performance and near-linear speedup of WU-UCT.

### 5.1 EXPERIMENTS ON THE "JOY CITY" GAME

Joy City is a level-oriented game with diverse and challenging gameplay. Players tap to eliminate connected items on the game board. To pass a level, players have to complete certain goals within a given number of steps.[5] The number of steps used to pass a level (termed game step) is the main performance metric, which differentiates masters from beginners. It is a challenging reinforcement learning task due to its large number of game-state (over $12^{9\times9}$) and high randomness in the transition. The goal of the production system is to accurately predict the user pass-rate of different game-levels, providing useful and fast feedback for game design. Powered by WU-UCT, the system runs $16\times$ faster while accurately predicting user pass-rate (8.6% MAE). In this subsection, we concentrate our analysis on the speedup and performance of WU-UCT using two typical game-levels (Level-35 and Level-58)[6], and refer the readers interested in the user pass-rate prediction system to Appendix C.

We evaluate WU-UCT with different numbers of expansion and simulation workers (from 1 to 16) and report the speedup results in Figures 4(a)–(b). For all experiments, we fix the total number of

---

[4]We refer the readers to Chaslot et al. (2008) for more details. The pseudo-code of the three algorithms is given in Appendix B. LeafP: Algorithm 4, TreeP: Algorithm 5, RootP: Algorithm 6.

[5]We refer it as the *tap game* below. See Appendix C.1 for more details about the game rules.

[6]Level-35 is relatively simple, requiring 18 steps for an average player to pass, while Level-58 is relatively difficult and needs more than 50 steps to solve.



(a) Speedup (level 35)  (b) Speedup (level 58)  (c) Performance (level 35)  (d) Performance (level 58)

Figure 4: WU-UCT speedup and performance. Results are averaged over 10 runs. WU-UCT achieves linear speedup with negligible performance loss (measured in game steps).

Table 1: The performance on 15 Atari games. Average episode return ($\pm$ standard deviation) over 10 trials are reported. The best average scores among parallel algorithms are highlighted in boldface. The mark * indicates that WU-UCT achieves statistically better performance ($p$-value $< 0.0011$ in paired t-test) than TreeP (no mark if both methods perform statistically similar). Similarly, the marks † and ‡ mean that WU-UCT performs statistically better than LeafP and RootP, respectively.

| Environment | WU-UCT | TreeP | LeafP | RootP | PPO | UCT |
|---|---|---|---|---|---|---|
| Alien | **5938**±1839 | 4200±1086 | 4280±1016 | 5206±282 | 1850 | 6820 |
| Boxing | **100**±0*†‡ | 99±0 | 95±4 | 98±1 | 94 | 100 |
| Breakout | **408**±21†‡ | 390±33 | 331±45 | 281±27 | 274 | 462 |
| Centipede | **1163034**±403910*†‡ | 439433±207601 | 162333±69575 | 184265±104405 | 4386 | 652810 |
| Freeway | **32**±0 | **32**±0 | 31±1 | **32**±0 | 32 | 32 |
| Gravitar | **5060**±568† | 4880±1162 | 3385±155 | 4160±1811 | 737 | 4900 |
| MsPacman | **19804**±2232*†‡ | 14000±2807 | 5378±685 | 7156±583 | 2096 | 23021 |
| NameThisGame | **29991**±1608* | 23326±2585 | 25390±3659 | 27440±9533 | 6254 | 38455 |
| RoadRunner | **46720**±1359*†‡ | 24680±3316 | 25452±2977 | 38300±1191 | 25076 | 52300 |
| Robotank | **101**±19 | 86±13 | 80±11 | 78±13 | 5 | 82 |
| Qbert | 13992±5596 | **14620**±5738 | 11655±5373 | 9465±3196 | 14293 | 17250 |
| SpaceInvaders | **3393**±292 | 2651±828 | 2435±1159 | 2543±809 | 942 | 3535 |
| Tennis | **4**±1*†‡ | -1±0 | -1±0 | 0±1 | -14 | 5 |
| TimePilot | **55130**±12474*† | 32600±2165 | 38075±2307 | 45100±7421 | 4342 | 52600 |
| Zaxxon | 39085±6838†‡ | **39579**±3942†‡ | 12300±821 | 13380±769 | 5008 | 46800 |

simulations to 500. First, note that when we have the same number of expansion workers and simulation workers, WU-UCT achieves linear speedup. Furthermore, Figures 4 also suggest that both the expansion workers and the simulation workers are crucial, since lowering the number of workers from either sets decreases the speedup. Besides the near-linear speedup property, WU-UCT suffers negligible performance loss with the increasing number of workers, as shown in Figures 4(c)–(d). The standard deviations of the performance (measured in the average game steps) over different numbers of expansion and simulation workers are only 0.67 and 1.22 for Level-35 and Level-58, respectively, which are much smaller than their average game steps (12 and 30).

## 5.2 EXPERIMENTS ON THE ATARI GAME BENCHMARK

We further evaluate WU-UCT on Atari Games (Bellemare et al., 2013), a classical benchmark for reinforcement learning (RL) and planning algorithms (Guo et al., 2014). The Atari Games are an ideal testbed for MCTS algorithms for its long planning horizon (several thousand), sparse reward, and complex game strategy. We compare WU-UCT to three parallel MCTS algorithms discussed in Section 4: *TreeP*, *LeafP*, and *RootP* (additional experiment results comparing WU-UCT with a variant of TreeP is provided in Appendix E). We also report the results of sequential UCT ($\approx 16\times$ slower than WU-UCT) and PPO (Schulman et al., 2017) as reference. Generally, the performance of sequential UCT sets an upper bound for parallel UCT algorithms. PPO is included since we used a distilled PPO policy network (Hinton et al., 2015; Rusu et al., 2015) as the roll-out policy for all other algorithms. It is considered as a performance lower bound for both parallel and sequential UCT algorithms. All experiments are performed with a total of 128 simulation steps, and all parallel algorithms use 16 workers (see Appendix D for the details).

We first compare the performance, measured by average episode reward, between WU-UCT and the baselines on 15 Atari games, which is done with 16 simulation workers and 1 expansion worker (for a fair comparison, since baselines do not parallel the expansion step). Each task is repeated 10 times with the mean and standard deviation reported in Table 1. Due to the better exploration-exploitation tradeoff during selection, WU-UCT out-performs all other parallel algorithms in 12 out of 15 tasks. Pairwise student t-test further show that WU-UCT performs significantly better (adjusted by the Bonferroni method, $p$-value $< 0.0011$) than TreeP, LeafP, and RootP in 7, 9, and 7 tasks, respectively. Next, we examine the influence of the number of simulation workers on the speed and the

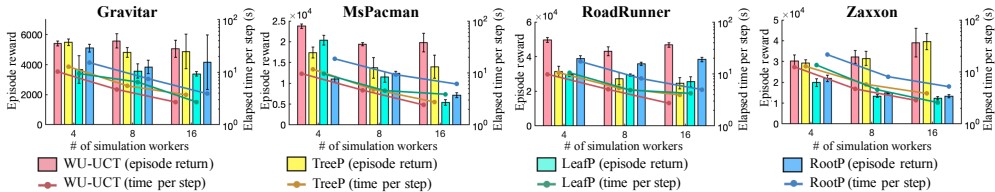

Figure 5: Speed and performance test of our WU-UCT along with three baselines on four Atari games. All experiments are repeated three times and the mean and standard deviation (for episode reward only) are reported. For WU-UCT, the number of expansion workers is fixed to be one.

performance. In Figure 5, we compare the average episode return as well as time consumption (per step) for 4, 8, and 16 simulation workers. The bar plots indicate that WU-UCT experiences little performance loss with an increasing number of workers, while the baselines exhibit significant performance degradation when heavily parallelized. WU-UCT also achieves the fastest speed compared to the baselines, thanks to the efficient master-worker architecture (Section 3.2). In conclusion, our proposed WU-UCT not only out-performs baseline approaches significantly under the same number of workers but also achieves negligible performance loss with the increasing level of parallelization.

## 6 RELATED WORK

**MCTS** Monte Carlo Tree Search is a planning method for optimal decision making in problems with either deterministic (Silver et al., 2016) or stochastic (Schäfer et al., 2008) environments. It has made a profound influence on Artificial Intelligence applications (Browne et al., 2012), and has even been applied to predict and mimic human behavior (van Opheusden et al., 2016). Recently, there has been a wide range of work combining MCTS and other learning methods, providing mutual improvements to both methods. For example, Guo et al. (2014) harnesses the power of MCTS to boost the performance of model-free RL approaches; Shen et al. (2018) bridges the gap between MCTS and graph-based search, outperforming RL and knowledge base completion baselines.

**Parallel MCTS** Many approaches have been developed to parallelize MCTS methods, with the objective being two-fold: achieve near-linear speedup under a large number of workers while maintaining the algorithm performance. Popular parallelization approaches of MCTS include leaf parallelization, root parallelization, and tree parallelization (Chaslot et al., 2008). Leaf parallelization aims at collecting better statistics by assigning multiple workers to query the same node (Cazenave & Jouandeau, 2007). However, this comes at the cost of wasting diversity of the tree search. Therefore, its performance degrades significantly despite the near-ideal speedup with the help of a client-server network architecture (Kato & Takeuchi, 2010). In root parallelization, multiple search trees are built and assigned to different workers. Additional work incorporates periodical synchronization of statistics from different trees, which results in better performance in real-world tasks (Bourki et al., 2010). However, a case study on Go reveals its inferiority with even a small number of workers (Soejima et al., 2010). On the other hand, tree parallelization uses multiple workers to traverse, perform queries, and update on a shared search tree. It benefits significantly from two techniques. First, a virtual loss is added to avoid querying the same node by different workers (Chaslot et al., 2008). This has been adopted in various successful applications of MCTS such as Go (Silver et al., 2016) and Dou-di-zhu (Whitehouse et al., 2011). Additionally, architecture side improvements such as using pipeline (Mirsoleimani et al., 2018b) or lock-free structure (Mirsoleimani et al., 2018a) speedup the algorithm significantly. However, though being able to increase diversity, virtual loss degrades the performance under even four workers (Mirsoleimani et al., 2017; Bourki et al., 2010). Finally, the idea of counting the unobserved samples to adjust the confidence interval in arm selection also appeared in Zhong et al. (2017). However, it mainly focuses on parallelizing the *multi-armed thresholding bandit problem* (Chen et al., 2014) instead of the tree search problem as we do.

## 7 CONCLUSION

This paper proposes WU-UCT, a novel parallel MCTS algorithm that addresses the problem of outdated statistics during parallelization by watching the number of unobserved samples. Based on the newly devised statistics, it modifies the UCT node-selection policy in a principled manner, which achieves effective exploration-exploitation tradeoff. Together with our efficiency-oriented system implementation, WU-UCT achieves near-optimal linear speedup with only limited performance loss across a wide range of tasks, including a deployed production system and Atari games.

## 8   ACKNOWLEDGEMENTS

This work is supported by Tencent AI Lab and Seattle AI Lab, Kwai Inc. We thank Xiangru Lian for his help on the system implementation.

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

# SUPPLEMENTARY MATERIAL

## A   ALGORITHM DETAILS FOR WU-UCT

The pseudo-code of WU-UCT is provided in Algorithm 1. Specifically, it provides the workflow of the master process. When the number of completed updates ($t_{complete}$) has not exceeded the maximum simulation step $T_{max}$ (a pre-defined hyperparameter), the main process repeatedly performs a modified *rollout* that consists of the following steps: *selection*, *expansion*, *simulation*, and *backpropagation*. The selection and backpropagation steps are performed in the main process, while the two others are assigned to the workers. The backpropagation step is divided into two sub-routines *incomplete update* (Algorithm 2) and *complete update* (Algorithm 3). The former is executed before simulation starts, while the latter is called after receiving simulation results. Task index $\tau$ is added to help the main process to track different tasks returned from the workers. To maximize efficiency, the master process keeps assigning expansion and simulation tasks until all workers are fully occupied.

**Communication overhead of WU-UCT**   The choice for centralized game-state storage stems from the following observations: (i) size of the game-state is usually small, which allows efficient inter-process transformation, and (ii) each game-state is used at most $|\mathcal{A}| + 1$ times,[7] thus is inefficient to store in multiple processes. Although this design may not be ideal for all tasks, it is at least a reasonable choice. During rollouts, game-states generated by any expansion worker may be later used by any other expansion and simulation workers. Therefore, either a per-task transformation or decentralized storage is needed. For the latter case, however, since a game-state will be used at most $|\mathcal{A}| + 1$ times, most workers will not need it, which results in inefficiency of the decentralized storage.

Another possible solution is to store the game-states in shared memory. However, to receive benefit from it, the following conditions should be satisfied: (i) each process can access (read/write) the memory relatively fast even if some collisions may happen, and (ii) the shared memory is big enough to hold all game-states that may be accessed. If the two conditions hold, we may be able to reduce the communication overhead. Since the communication overhead is negligible even with 16 simulation and expansion workers (as shown in Figures 2(b) and 2(c)), we should consider using more workers to speedup the algorithm.

## B   ALGORITHM OVERVIEW OF BASELINE APPROACHES

We give an overview of three baseline parallel UCT algorithms: Leaf Parallelization (LeafP), Tree Parallelization (TreeP) with virtual loss, and Root Parallelization (RootP), with the objective of providing a comprehensive view to the readers. We refer readers interested in the details of these algorithms to Chaslot et al. (2008). As suggested by their names, LeafP, TreeP, and RootP parallelized different parts of the search tree. Specifically, LeafP (Algorithm 4) parallelizes only the simulation process: whenever a node (state) is selected to query, all workers perform simulations individually to evaluate it. The main process (master) then waits for all workers to complete simulation and return their respective cumulative rewards, which are used to update the traversed nodes' statistics.

TreeP (Algorithm 5) parallelizes the whole tree search algorithm by allowing different workers to access a shared search tree simultaneously. Each worker individually performs the selection, expansion, simulation, and back-propagation steps and update the nodes' statistics. To discourage querying the same node, individual workers subtract a virtual loss $r_{VL}$ ($r_{VL}$ is a hyper-parameter of the algorithm) to each of its traversed node during the selection process, and add it back ($+r_{VL}$) during back-propagation. This allows nodes currently being evaluated by some workers to have lower utility scores (4) and will be less likely to be chosen by other workers, which improves the diversity of the node visited by different workers simultaneously.

Silver et al. (2017) and Segal (2010) introduced an additional way to add pseudo reward into the traversed nodes. See Appendix E for details of this variant of TreeP and more experiments of it on Atari games.

---

[7]In our setup, the game state will only be used for 1 time to start simulation and $|\mathcal{A}|$ times to initialize expansion.

---

**Algorithm 1** WU-UCT

---

**Input:** environment emulator $\mathcal{E}$, root tree node $s_{root}$, maximum simulation step $T_{max}$, maximum simulation depth $d_{max}$, number of expansion workers $N_{exp}$, and number of simulation workers $N_{sim}$

**Initialize:** expansion worker pool $\mathcal{W}_{exp}$, simulation worker pool $\mathcal{W}_{sim}$, game-state buffer $\mathcal{B}$, $t \leftarrow 0$, and $t_{complete} \leftarrow 0$

**while** $t_{complete} < T_{max}$ **do**

    Traverse the tree top down from root node $s_{root}$ following (4) until (i) its depth greater than $d_{max}$, (ii) it is a leaf node, or (iii) it is a node that has not been fully expanded and *random()* $< 0.5$

    **if** expansion is required **then**

        $\bar{s} \leftarrow$ shallow copy of the current node

        Assign expansion task $(t, \bar{s})$ to pool $\mathcal{W}_{exp}$ **//** $t$ **is the task index**

    **else**

        assign simulation task $(t, s)$ to pool $\mathcal{W}_{sim}$ if episode not terminated

        Call **incomplete_update**$(s)$; if episode terminated, call **complete_update**$(t, s, 0.0)$

    **end if**

    **if** $\mathcal{W}_{exp}$ fully occupied **then**

        Wait for a expansion task with return: (task index $\tau$, game state $s$, reward $r$, terminal signal $d$, task index $\tau$); expand the tree according to $s$, $\tau$, $r$, and $d$; assign simulation task $(\tau, s)$ to pool $\mathcal{W}_{sim}$

        Call **incomplete_update**$(t, s)$

    **else continue**

    **if** $\mathcal{W}_{sim}$ fully occupied **then**

        Wait for a simulation task with return: (task index $\tau$, node $s$, cumulative reward $\bar{r}$)

        Call **complete_update**$(\tau, s, \bar{r})$; $t_{complete} \leftarrow t_{complete} + 1$

    **else continue**

    $t \leftarrow t + 1$

**end while**

---

**Algorithm 2** incomplete_update

---

**input:** node $s$

**while** $n \neq$ null **do**

    $O_s \leftarrow O_s + 1$

    $s \leftarrow \mathcal{PR}(s)$ **//** $\mathcal{PR}(s)$ **denotes the parent node of** $s$

**end while**

---

**Algorithm 3** complete_update

---

**input:** task index $t$, node $s$, reward $\bar{r}$

**while** $n \neq$ null **do**

    $N_s \leftarrow N_s + 1$; $O_s \leftarrow O_s - 1$

    Retrieve reward $r$ according to task index $t$

    $\bar{r} \leftarrow r + \gamma\bar{r}$; $V_s \leftarrow \frac{N_s - 1}{N_s}V_s + \frac{1}{N_s}\bar{r}$

    $s \leftarrow \mathcal{PR}(s)$ **//** $\mathcal{PR}(s)$ **denotes the parent node of** $s$

**end while**

---

As hinted by its name, RootP (Algorithm 6) parallelizes the root node. Specifically, in an initialization step, all children of the root node is expanded, and different workers are assigned to perform rollouts using the expanded child nodes as the root node of the search tree. The algorithm evenly distribute the workload such that the number of rollouts starting from all child nodes is $T_{max}/M$, where $M$ is the number of workers. After the job assignment, all workers construct search trees in their own local memories and perform sequential tree search until their assigned tasks are finished. Finally, the main process collects statistics from all workers and return the predicted best action of the state represented by the root node of the search tree.

## C   EXPERIMENT DETAILS AND SYSTEM DESCRIPTION OF THE JOY CITY TASK

This section describes the basic rules of the Joy City game (Appendix C.1) as well as details about the deployed user pass-rate prediction system (Appendix C.2).

### C.1   DESCRIPTION OF THE JOY CITY GAME

This section serves as an introduction to the basic rules of the tap game. Figure 7 depicts several screenshots of the game. In the main frame, there is a $9 \times 9$ grid, where each cell contains an item. We can click cells with connected color regions to eliminate them (i.e., if the cell represented by the purple dot in the first screenshot of Figure 6(a) is tapped, the region contains blue boxes will be "eliminated"). The remaining cells then collapse to fill in the gaps of exploded ones. To goal is to fulfill all level requirements (goals) within a fixed number of clicks. Figure 6(a) provides consecutive snapshots for playing level 10 of the game. The goal of this level is depicted on the

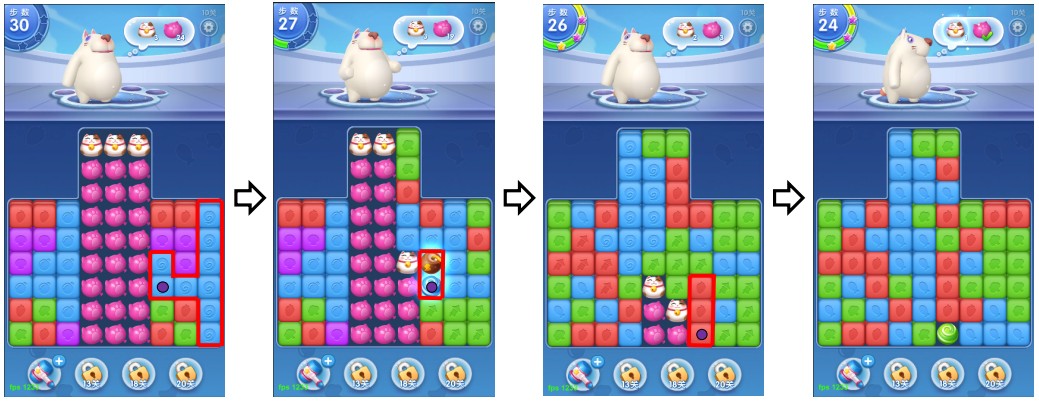

(a) A demonstrated game play in level 10 of the tap game. Purple dots refers to the tapped cell, and red regions indicate directly eliminated cells.

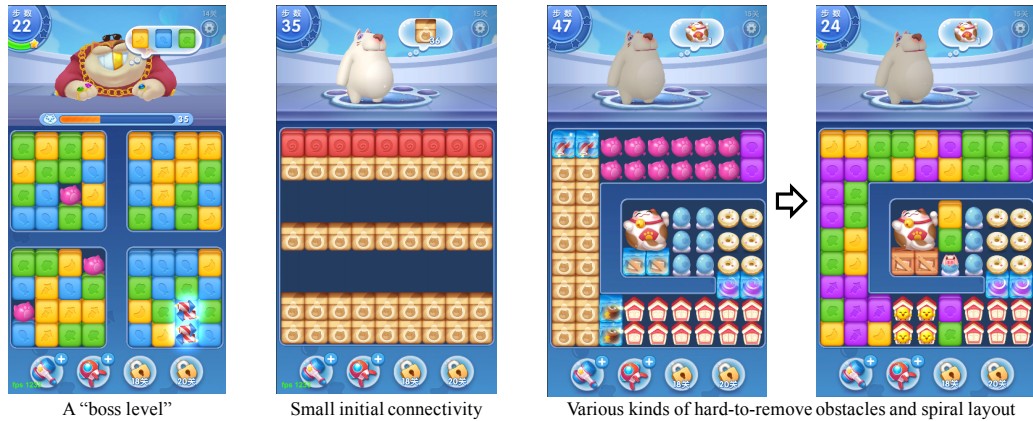

A "boss level"          Small initial connectivity          Various kinds of hard-to-remove obstacles and spiral layout

(b) Examples of levels with different rule, difficulty, and layout.

Figure 6: Snapshots of the Tap-elimination game.

top, which is 3 "cats" and 24 "balloons". The top-left corner represents the number of remaining steps. Players have to accomplish all given goals before the step runs out. Figure 6(a) demonstrates successful gameplay, where only 6 steps are used to complete the level. In each of the three left frames, the cell noted by the purple circle is clicked. Immediately, the same-color region marked with a red frame is eliminated. Different goal objects/obstacle objects react differently. For instance, when some cell is exploded beside a balloon, it will also explode. Frame two demonstrates the use of props. Tapping regions with connectivity above a certain threshold will provide prop as a bonus. They have special effects that can help players pass the level faster. Finally, in the last screenshot, all goals are completed and we pass the level.

Figure 6(b) further demonstrates the variety of levels. Specifically, the left-most frame depicts a special "boss level", where the goal is the "defeat" the evil cat. The cat will randomly throw objects to the cells, adding additional randomness. Three other frames illustrate relatively hard levels, which is revealed from their low-connectivity, abundance and complexity of the obstacles, and special layout.

## C.2    DETAILS OF THE LEVEL PASS-RATE PREDICTION SYSTEM

During a game design cycle, to achieve the desired game pass-rates, a game designer needs to hire many human testers to extensively test

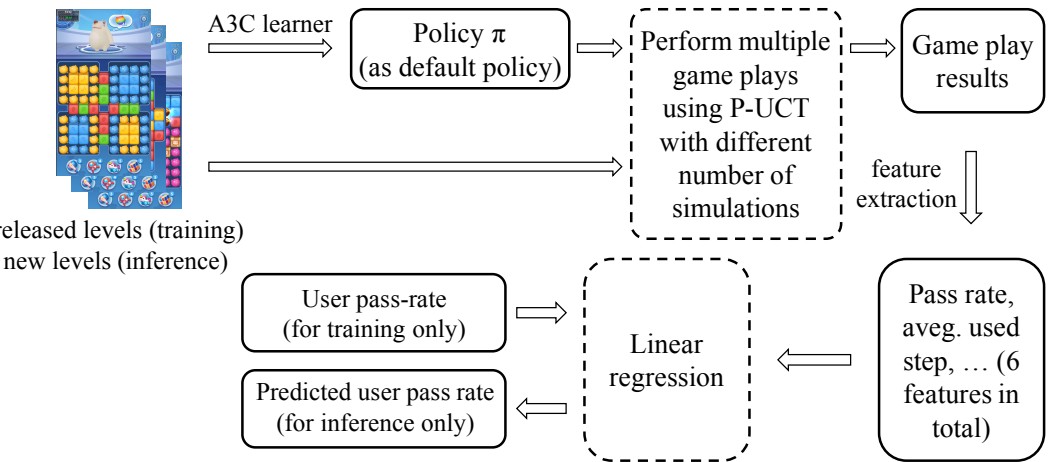

Figure 7: Our deployed user pass-rate prediction system.

all the levels before its release, which generally takes a long time and is inaccurate. Therefore, it would greatly reduce the game design cycle if we can develop a testing system that is able to provide *quick* and *accurate* feedback about the user pass-rates. Figure 7 gives an overview of our deployed user pass-rate prediction system, where WU-UCT is used to mimic average user performance and provide features for predicting the human pass-rate. As we have shown in the main paper, it can achieve significant speedup without significant performance loss,[8] allowing the game designer to get the feedback in 20 minutes instead of 12 hours. Specifically, we use WU-UCT with different numbers of rollouts to mimic players with different skill levels, where WU-UCT with 10 rollouts is used to represent average players while the agent with 100 rollouts mimics skillful players. This is verified by the pair-wise sample t-test result provided in Table 2. With 10 simulations, the WU-UCT agent performs statistically similar ($p$-value $> 5\%$) to human players, while with 100-simulation, the agent performs statistically better ($p$-value $< 5\%$). Besides, Figure 8 shows that our pass-rate prediction system achieves 8.6% mean absolute error (MAE) on 130 released game-levels, with 93% of them having MAE less than 20%.

The system consists of two working phases, i.e., training and inference. Specifically, training and validation are done on 300 levels that have been released in a test version of the game. In the training phase, the system has access to both the level and players' pass-rate, while only levels are available in the inference phase, and the system needs to give quick and accurate feedback about the (predicted) user pass-rate. In both phases, the levels are first fed into an asynchronous advantage actor-critic (A3C) (Mnih et al., 2016) learner for a base policy $\pi$. It is then used by the WU-UCT agent as a prior to select expand action as well as the default policy for simulation. We then use WU-UCT to perform multiple gameplays. The maximum depth and width (maximum number of child nodes for each node) of the search tree is 10 and 5, respectively. The number of simulations is set to 10 and 100 to get AI bots with different skill levels. Six features (three for both the 10-simulation and 100-simulation agent) are extracted from the gameplay results. Specifically, the features are AI's pass-rate, average used step divided by the provided step (the number at the top-left corner in the screenshots in Figure 6), and median used step divided by the provided step. During training, the features, as well as the players' pass-rate, is used to learn a linear regressor, while in the inference phase, the regression model is used to predict user pass-rate.

## C.3 ADDITIONAL EXPERIMENTAL RESULTS

In this section, we list the additional experimental results. In Table 3, we report the specific speedup number for different numbers of expansion worker and simulation workers.

---

[8]Due to the complexity the tap game, model-free RL algorithms such as A3C (Mnih et al., 2016) and PPO (Schulman et al., 2017) fail to achieve satisfactory performance and thus cannot perform an accurate prediction. On the other hand, MCTS could achieve good performance but takes a long time in testing.

Table 2: Pair-wise sample t-test of pass-rate across 130 levels between two AI bots (different number of MCTS rollouts) and the players. "Avg. diff" means the average difference between the pass-rate of the bot and that of the human players. $p$-value measures the likelihood that two sets of paired samples are statistically similar (i.e. larger means similar). Effect size measures the strength of the difference (larger means greater difference).

| AI bot | # rollouts | Avg. diff. | Effect size | $p$-value |
|--------|-----------|-----------|-------------|-----------|
| WU-UCT | 10 | -1.54 | 0.07 | 0.4120 |
| WU-UCT | 100 | 22.18 | 0.88 | 0.0000 |

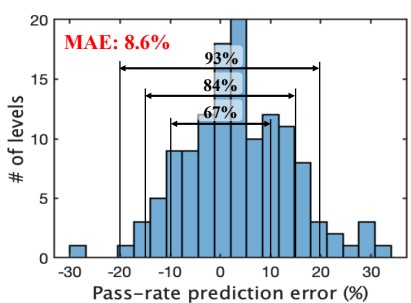

Figure 8: Distribution of the pass-rate prediction error on 130 game-levels.

Table 3: Speedup on two levels of the tap game. $M_e$ is the number of expansion workers and $M_s$ is the number of simulation workers.

| Lv. | Level 35 | | | | | Level 58 | | | | |
|-----|------|-----|-----|------|------|-----|-----|-----|------|------|
| $M_s$ / $M_e$ | 1 | 2 | 4 | 8 | 16 | 1 | 2 | 4 | 8 | 16 |
| 1 | 1.0 | 2.0 | 2.8 | 3.6 | 4.5 | 1.0 | 1.8 | 4.1 | 4.8 | 5.1 |
| 2 | 1.4 | 2.2 | 4.1 | 5.7 | 6.3 | 1.1 | 3.1 | 5.3 | 6.7 | 8.4 |
| 4 | 1.7 | 2.5 | 4.5 | 8.4 | 8.8 | 1.1 | 3.4 | 6.1 | 10.1 | 12.8 |
| 8 | 2.3 | 3.0 | 5.1 | 10.1 | 12.8 | 1.2 | 3.6 | 6.7 | 13.2 | 16.1 |
| 16 | 2.9 | 3.7 | 5.7 | 11.2 | 15.5 | 1.2 | 3.8 | 7.6 | 16.1 | 20.9 |

# D EXPERIMENT DETAILS OF THE ATARI GAMES

This section provides the implementation details of the experiments on Atari games. Specifically, we first describe the training pipeline of the default policy. We then illustrate how the default policy is connected with MCTS algorithm to perform simulation.

**Training default policy for MCTS** To allow better overall performance, we used the Proximal Policy Gradient (PPO) (Schulman et al., 2017), one of the state-of-the-art on-policy model-free reinforcement learning (RL) algorithms. We adopted the highest-starred third-party code of PPO on GitHub. The implementation uses the same hyper-parameters with the original paper. The architecture of the policy network is shown in Figure 9. The original PPO network is trained on 10 million frames for each task. To reduce computation count, we reduce the network size using network

Table 4: Performance of the original PPO policy and our distilled policy on 15 Atari games.

| Environment | Origin PPO policy | Distilled policy |
|-------------|-------------------|------------------|
| Alien | 1850 | 850 |
| Boxing | 94 | 7 |
| Breakout | 274 | 191 |
| Centipede | 4386 | 1701 |
| Freeway | 32 | 32 |
| Gravitar | 737 | 600 |
| MsPacman | 2096 | 1860 |
| NameThisGame | 6254 | 6354 |
| RoadRunner | 25076 | 26600 |
| Robotank | 5 | 13 |
| Qbert | 14293 | 12725 |
| SpaceInvaders | 942 | 1015 |
| Tennis | -14 | -10 |
| TimePilot | 4342 | 4400 |
| Zaxxon | 5008 | 3504 |

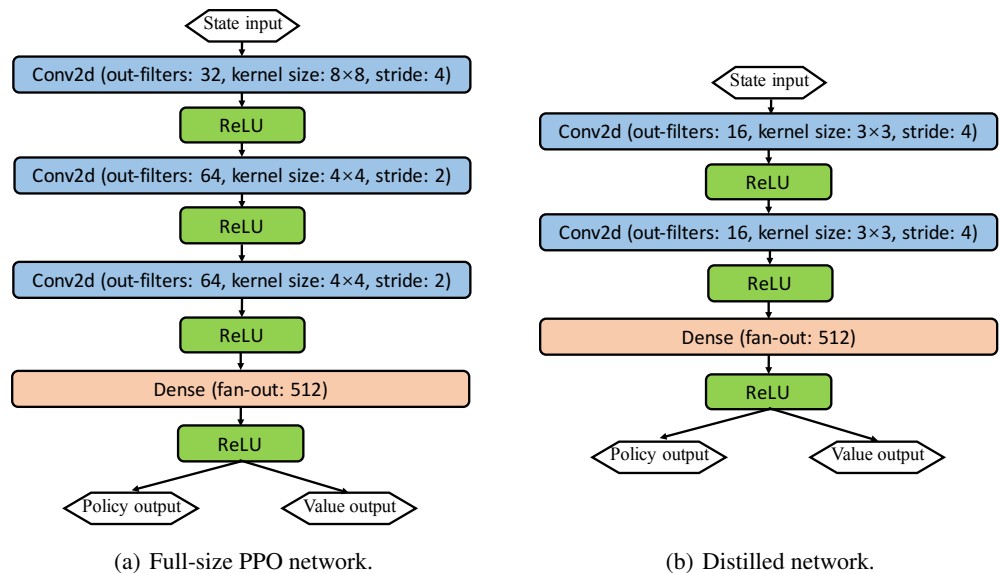

(a) Full-size PPO network.  (b) Distilled network.

Figure 9: Architecture of the original PPO network (left) and the distilled network (right).

distillation (Hinton et al., 2015). Specifically, it is a teacher-student training framework where the student (distilled) network mimics the output of the teacher network. Samples are collected by the PPO network with the $\epsilon$-greedy strategy ($\epsilon = 0.1$). The student network optimizes its parameters to minimize the mean square error of the policy's logits as well as the value. Performance of the original PPO policy network as well as the distilled network is provided in Table 4.

**MCTS simulation** Both the policy output and the value output of the distilled network is used in the simulation phase. Particularly, if a simulation is started from state $s$, rollout is performed using the policy network with an upper bound of 100 steps and reaches the leaf state $s'$. If the environment does not terminate, the full return is computed by the intermediate rewards plus the value function at state $s'$. Formally, the cumulative reward provided by the simulation is $R_{simu} = \sum_{i=0}^{99} \gamma^i r_i + \gamma^{100} V(s')$, where $V(s)$ denotes the value of state $s$. To reduce the variance of Monte Carlo sampling, we average it with the value function $V(s)$ at state $s$. The final simulation return is then $R = 0.5 R_{simu} + 0.5 V(s)$.

**Hyperparameters and experiment details for WU-UCT** For all tree search based algorithms (i.e., WU-UCT, TreeP, LeafP, and RootP), the maximum depth of the search tree is set to 100. The search width is limited by 20 and the maximum number of simulations is 128. The discount factor $\gamma$ is set to 0.99 (note that the reported score is not discounted). When performing gameplays, a tree search subroutine is called to plan for the best action in each time step. The sub-routine iteratively constructs a search tree from its initialization with a root node only. Experiments are deployed on 4 Intel® Xeon® E5-2650 v4 CPUs and 8 NVIDIA® GeForce® RTX 2080 Ti GPUs. To minimize the speed fluctuation caused by different workloads on the machine, we ensure that the total number of simulation workers is smaller than the total number of CPU cores, allowing each process to fully occupy each single core. The WU-UCT is implemented with multiple processes, with an inter-process pipe between the master process and each worker process.

**Hyperparameters and experiments for baseline algorithms** Being unable to find appropriate third-party packages for baseline algorithms (i.e., tree parallelization, leaf parallelization, and root parallelization), we built our implementation of them based on the corresponding papers. Building all algorithms in the same package additionally allows us to accurately conduct speed-tests as it elim-

---

[10]The task "Tennis" is not included in the calculation of the average percentile improvement due to the average episode return 0 in RootP.

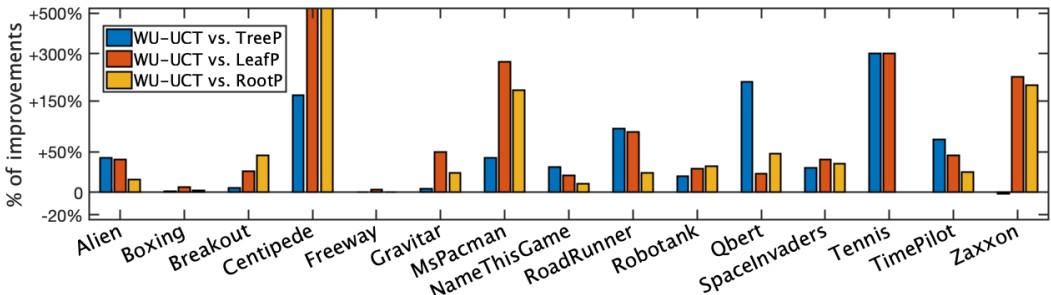

Figure 10: Relative performance between WU-UCT and three baseline approaches on 15 Atari benchmarks. Relative performance is calculated according to the mean episode reward in 3 trials. The average percentile improvement of WU-UCT on TreeP, LeafP, and RootP is 49%, 104%, and 82%, respectively.[10]

Table 5: Comparison between WU-UCT and three TreeP variants on 12 Atari games. Average episode return ($\pm$ standard deviation) over 10 runs are reported. The best average scores among listed algorithms are highlighted in boldface. Hyper-parameter $r_{VL}$ refers to the virtual loss added before each simulation starts, and $n_{VL}$ similarly denotes the number of virtual visit counts added before each simulation starts.

| Environment | WU-UCT | TreeP $(r_{VL} = n_{VL} = 1)$ | TreeP $(r_{VL} = n_{VL} = 2)$ | TreeP $(r_{VL} = n_{VL} = 3)$ |
|---|---|---|---|---|
| Alien | **5938**$\pm$1839 | 4850$\pm$357 | 4935$\pm$60 | 5000$\pm$0 |
| Boxing | **100**$\pm$0 | 99$\pm$1 | 99$\pm$0 | 99$\pm$1 |
| Breakout | 408$\pm$21 | 379$\pm$43 | 265$\pm$50 | **463**$\pm$60 |
| Freeway | **32**$\pm$0 | **32**$\pm$0 | **32**$\pm$0 | **32**$\pm$0 |
| Gravitar | **5060**$\pm$568 | 3500$\pm$707 | 4105$\pm$463 | 4950$\pm$141 |
| MsPacman | **19804**$\pm$2232 | 13160$\pm$462 | 12991$\pm$851 | 8640$\pm$438 |
| RoadRunner | **46720**$\pm$1359 | 29800$\pm$282 | 28550$\pm$459 | 29400$\pm$494 |
| Qbert | 13992$\pm$5596 | **17055**$\pm$353 | 13425$\pm$194 | 9075$\pm$53 |
| SpaceInvaders | **3393**$\pm$292 | 2305$\pm$176 | 3210$\pm$127 | 3020$\pm$42 |
| Tennis | **4**$\pm$1 | 1$\pm$0 | 1$\pm$0 | 1$\pm$0 |
| TimePilot | 55130$\pm$12474 | **52500**$\pm$707 | 49800$\pm$212 | 32400$\pm$1697 |
| Zaxxon | **39085**$\pm$6838 | 24300$\pm$2828 | 24600$\pm$424 | 37550$\pm$1096 |

inates other factors (e.g. different language) that may bias the result. Specifically, leaf parallelization is implemented with a master-worker structure: when the main process enters the simulation step, it assigns the task to all workers. When return from all workers is available, the master process performs backpropagation according to these statistics and begin a new rollout.

As suggested by Browne et al. (2012), tree parallelization is implemented using a decentralized structure, i.e., each worker performs rollouts on a shared search tree. At the selection step, each traversed node is added a fixed virtual loss $-r_{VL}$ to guarantee diversity of the tree search. When performing backpropagation, $r_{VL}$ is added back to the traversed nodes. $r_{VL}$ is chosen from 1.0 and 5.0 for each particular task. In other words, we ran TreeP with $r_{VL} = 1.0$ and $r_{VL} = 5.0$ for each task, and report the better result.

Root parallelization is implemented according to Chaslot et al. (2008). Similar to leaf parallelization, root parallelization consists of sub-processes that do not share information with each other. At the beginning of the tree search process, each sub-process is assigned several actions of the root node to query. They then perform sequential UCT rollouts until reaches a pre-defined maximum number of rollouts. When all sub-processes complete the jobs, statistics from them are gathered by the main process, and are used to choose the best action.

# E  ADDITIONAL EXPERIMENTS ON THE ATARI GAMES

This section provides additional experiment results to compare WU-UCT with another variant of the Tree Parallelization (TreeP) algorithm. As suggested by Silver et al. (2016), besides pre-adjusting the value $V$ with virtual loss $r_{VL}$, pre-adjusted visit count can also be used to penalize $V$. In this variant of TreeP, both the virtual loss $r_{VL}$ and a hand-crafted count correction $n_{VL}$ (termed the virtual pseudo-count) is added to adjust $V$. Specifically, the value of node $s$ is adjusted as

$$V_s' \overset{def}{=} \frac{N_s V_s - r_{VL}}{N_s + n_{VL}}, \tag{7}$$

which is used in the UCT selection phase. Table 5 compares WU-UCT with this TreeP variant using both virtual loss and virtual pseudo-count (i.e., Eq. 7). Three sets of hyper-parameters are used in TreeP, which are described in the caption of the table (i.e., $r_{VL} = n_{VL} = 1$, $r_{VL} = n_{VL} = 2$, and $r_{VL} = n_{VL} = 3$). All other experiment setups are the same as Section 5.2 and Appendix D. Table 5 indicates that on 9 out of 12 tasks, WU-UCT out-performs this new baseline (with its best hyper-parameters). Furthermore, we also observe that TreeP does not have an optimal set of hyper-parameters that performs uniformly well on all tasks. In other words, to perform well, TreeP needs to conduct per-task hyper-parameter tuning. On the other hand, WU-UCT performs consistently well across different tasks.

Conceptually, WU-UCT is designed based on the fact that on-going simulations (unobserved samples) will eventually return the results, so their number should be tracked and used to adaptively adjust the UCT selection process. On the other hand, TreeP uses artificially designed virtual loss $r_{VL}$ and virtual pseudo-count $n_{VL}$ to discourage other threads from simultaneously exploring the same node. Therefore, WU-UCT achieves a better exploration-exploitation tradeoff in parallelization, which leads to better performance as confirmed by the experimental results given in Table 5.

---

**Algorithm 4** Leaf Parallelization (LeafP)

---

**Input:** environment emulator $\mathcal{E}$, prior policy $\pi$, root tree node $s_{root}$, maximum simulation step $T_{max}$, maximum simulation depth $d_{max}$, and number of workers $N_{sim}$
**Initialize:** $t_{complete} \leftarrow 0$
**while** $t_{complete} < T_{max}$ **do**
    Traverse the tree top down from root node $s_{root}$ following (2) until (i) its depth greater than $d_{max}$, (ii) it is a leaf node, or (iii) it is a node that has not been fully expanded and *random()* < 0.5
    $s' \leftarrow$ **expand**$(s, \mathcal{E}, \pi)$
    Each of the simulation workers perform roll-out beginning from $s'$
    Wait until all workers completed simulation and returned cumulative reward $\{\bar{r}_i\}_{i=1}^{N_{sim}}$ ($\bar{r}_i$ is returned by worker $i$)
    **for** $i = 1 : N_{sim}$ **do**
        Call **back_propagation**$(s, \bar{r}_i)$
    **end for**
    $t_{complete} \leftarrow t_{complete} + N_{sim}$
**end while**

---

**Algorithm 5** Tree Parallelization (TreeP)

---

**Input:** environment emulator $\mathcal{E}$, prior policy $\pi$, root tree node $s_{root}$, virtual loss $r_{VL}$, maximum simulation step $T_{max}$, maximum simulation depth $d_{max}$, and number of workers $N_{sim}$
**Initialize:** $t_{complete} \leftarrow 0$
**Initialize:** $N_{sim}$ processes, each with access to the environment emulator, the prior policy, and the search tree
**Perform asynchronously** in each of the $N_{sim}$ workers
    Traverse the tree top down from root node $s_{root}$ following (2) until (i) its depth greater than $d_{max}$, (ii) it is a leaf node, or (iii) it is a node that has not been fully expanded and *random()* < 0.5
    Add virtual loss to each of the traversed node: $V_s \leftarrow V_s - r_{VL}$ for each traversed $s$
    $s' \leftarrow$ **expand**$(s, \mathcal{E}, \pi)$
    Perform roll-out beginning from $s'$
    $\bar{r} \leftarrow$ the returned cumulative reward of the roll-out
    Call **back_propagation**$(s, \bar{r}_i)$
    Remove virtual loss from each of the traversed node: $V_s \leftarrow V_s + r_{VL}$ for each traversed $s$
    $t_{complete} \leftarrow t_{complete} + 1$
    **if** $t_{complete} \geq T_{max}$
        Terminate current process
    **end if**
**end**

---

**Algorithm 6** Root Parallelization (RootP)

---

**Input:** environment emulator $\mathcal{E}$, prior policy $\pi$, root tree node $s_{root}$, maximum simulation step $T_{max}$, maximum simulation depth $d_{max}$, and number of workers $N_{sim}$
**Initialize:** $t_{complete,i} \leftarrow 0$ for $i = 1 : N_{sim}$
**Initialize:** $N_{sim}$ processes, each with access to the environment emulator, the prior policy, and the search tree
Expand all child nodes of $s_{root}$
$T_{avg} \leftarrow ceil(T_{max}/|\mathcal{A}|)$ ($|\mathcal{A}|$ is the number of actions)
Averagely distribute the workload (perform tree search $T_{avg}$ times on each child of $s_{root}$) to the $N_{sim}$ workers, and copy the corresponding child nodes to the worker's local memory.
**Perform asynchronously** in each of the $N_{sim}$ workers ($i$ denotes the thread ID)
    $s_{root,i} \leftarrow$ Select a child of $s_{root}$ according to its allocated budget
    Traverse the tree top down from root node $s_{root,i}$ following (2) until (i) its depth greater than $d_{max}$, (ii) it is a leaf node, or (iii) it is a node that has not been fully expanded and *random()* < 0.5
    Add virtual loss to each of the traversed node: $V_s \leftarrow V_s - r_{VL}$ for each traversed $n$
    $s' \leftarrow$ **expand**$(s, \mathcal{E}, \pi)$
    Perform roll-out beginning from $s'$
    $\bar{r} \leftarrow$ the returned cumulative reward of the roll-out
    Call **back_propagation**$(s, \bar{r}_i)$
    Remove virtual loss from each of the traversed node: $V_s \leftarrow V_s + r_{VL}$ for each traversed $n$
    $t_{complete,i} \leftarrow t_{complete,i} + 1$
    **if** $t_{complete,i} \geq T_{avg}$
        Terminate current process
    **end if**
**end**
Gather child nodes' statistics from all workers

---

---

**Algorithm 7** expansion

---

**input:** node $s$, environment emulator $\mathcal{E}$, prior policy $\pi$
$a \leftarrow$ random action drawn from $\pi(\cdot \mid s)$
**while** $s$ has expanded $a$ **do**
   $a \leftarrow$ random action drawn from $\pi(\cdot \mid s)$
**end while**
$s', r, d \leftarrow$ performing $a$ in $s$ according to $\mathcal{E}$ ($d$: terminal signal)
$s' \leftarrow$ a new node constructed according to $s$
Store reward signal $r$ and termination indicator $d$ in $s'$
Link $s'$ as the child of $s$ by the node corresponding to $a$

---

**Algorithm 8** back_propagation

---

**input:** node $s$, cumulative reward $\bar{r}$
**while** $s \neq$ null **do**
   $N_s \leftarrow N_s + 1$
   Retrieve the reward $r$ in the current node $s$ (which is collected during its expansion)
   $\bar{r} \leftarrow r + \gamma \bar{r}$
   $V_s \leftarrow \frac{N_s - 1}{N_s} V_s + \frac{1}{N_s} \bar{r}$
   $s \leftarrow \mathcal{PR}(s)$ **//** $\mathcal{PR}(s)$ **denotes the parent node of** $s$
**end while**

---

