# OpenReview forum: "Watch the Unobserved: A Simple Approach to Parallelizing Monte Carlo Tree Search"
_ICLR.cc/2020/Conference — Accept (Talk)_

### Official Review · AnonReviewer2 · 2019-10-10
**Official Blind Review #2**

**Rating:** 8

**Review:**

This paper introduces a novel approach to parallelizing Monte Carlo Tree Search
which achieves speedups roughly linear in the number of parallel workers while
avoiding significant loss in performance. The key idea to the
approach is to keep additional statistics about the number of
on-going simulations from each of the nodes in the tree. The approach is
evaluated in terms of speed and performance on the Atari benchmark and in a
user pass-rate prediction task in a mobile game.

I recommend that this paper be accepted. The approach is well motivated and clearly
explained, and is supported by the experimental results. The experiments are reasonably thorough and
demonstrate the claims made in the paper. The paper itself is very well-written, and all-around
felt very polished. Overall I am enthusiastic about the paper and have only a few concerns, detailed below.

- I suggest doing more runs of the Atari experiment. Three runs of the experiment does not
seem large enough to make valid claims about statistical significance. This is especially
concerning because claims of statistical significance are made via t-testing, which assumes
that the data is normally distributed. Three runs is simply too few to be making conclusions
about statistical significance using t-testing. I think that this is a fair request to make and
could reasonably be done before the camera-ready deadline, if the paper is accepted.

- The experiments in Atari compare against a model-free Reinforcement Learning baseline, PPO.
Was there a set clock time that all methods had to adhere to? Or alternatively, was it verified that
PPO and the MCTS methods are afforded approximately equal computation time? If not, it seems
like the MCTS methods  could have an unfair advantage against PPO, especially if they are
allowed to take as long as  necessary to complete their rollouts. This computational bias
could potentially be remedied by  allowing PPO to use sufficiently complex function
approximators, or by setting the number of simulations used by the MCTS methods
such that their computation time is roughly equal to that of PPO.

- I would be careful about stating that PPO is a state-of-the-art baseline. State-of-the-art is a big claim, and I'm not quite sure that it's true for PPO. PPO's performance is typically only compared to other policy-based RL methods; it's hard to say that it's a state-of-the-art method when there's a lack of published work comparing it against the well-known value-based approaches, like Rainbow. I suggest softening the language there unless you're confident that PPO truly is considered a state-of-the-art baseline.

**Experience Assessment:**

I do not know much about this area.

**Review Assessment: Checking Correctness Of Derivations And Theory:**

N/A

**Review Assessment: Checking Correctness Of Experiments:**

I carefully checked the experiments.

**Review Assessment: Thoroughness In Paper Reading:**

I read the paper thoroughly.

---

> ### Author Response · Authors · 2019-11-10
> **Response to reviewer #2**
>
> Thank you for the valuable suggestions.
>
> First, to improve the statistical significance tests, we launched new experiments to perform 10 runs for each environment and each model in the Atari game task. So far, we have completed 5 Atari games (out of 15 games) and report the results below (and in the revised paper).  And we will post the results for all other on-going experiments once they are completed. We will also update Figure 5 and 10 (in the revised manuscript) after we finish the experiments.
>
> Table: Additional experimental results on 5 Atari games with 10 independent runs. (note: based on the suggestion by Reviewer #4, we have changed our algorithm name from P-UCT to WU-UCT to avoid potential confusion with an existing algorithm named PUCT. However, in the response below, we will still use P-UCT for your convenience.)
> +=============+===========+===========+===========+==========+
> + Environments +      P-UCT      +      TreeP        +     LeafP        +     RootP    +
> +=============+===========+===========+===========+==========+
> +      Freeway       +     32±0         +      32±0         +       31±1        +      32±0      +
> +=============+===========+===========+===========+==========+
> +      Gravitar       +   5060±568   +  4880±1162  +  3385±155     + 4160±1811 +
> +=============+===========+===========+===========+==========+
> +    MsPacman    + 19804±2232 + 14000±2807 +   5378±685   +  7156±583  +
> +=============+===========+===========+===========+==========+
> +  RoadRunner   + 46720±1359 + 24680±3316 + 25452±2977 +38300±1191+
> +=============+===========+===========+===========+==========+
> +      Zaxxon        + 39579±3942 + 38839±4128 +   12300±821  + 13380±769 +
> +=============+===========+===========+===========+==========+
>
> Second, we would like to clarify that the purpose of including PPO in Table 2 for the Atari experiments (Section 5.2) is to use it as the performance lower bound for all MCTS algorithms. Recall that we used distilled PPO policies (with network distillation) as the roll-out policy for all MCTS algorithms (i.e., P-UCT, TreeP, LeafP, and RootP), which is briefly described in Section 5.2 and detailed in Appendix D. Therefore, the performance of PPO is added here as a reference, which serves as a lower expected bound of UCT algorithms (both sequential and parallelized) since we expect them to perform significantly better than their roll-out policy. Our main focus for Table 2 is the relative performance between different parallel MCTS algorithms, including P-UCT. To avoid confusion, we revised the first paragraph as well as Table 2 to clarify the intention of including PPO.
>
> Third, in addition to the “performance lower bound” given by PPO, we also included the results of the sequential UCT in the revised manuscript (suggested by Reviewer #4), and use it as the performance upper bound for all parallel UCT algorithms. This is because, in general, we do not expect any parallel algorithm to outperform its sequential counterpart. These results empirically demonstrate the performance degradation caused by parallelizing UCT. It shows that our P-UCT has much smaller performance degradation compared to other methods.
>
> Finally, based on your feedback, we have revised the corresponding statement regarding PPO as “a state-of-the-art baseline”. We use PPO as our roll-out policy because it is actually a competitive model-free RL algorithm, which achieves reasonably well performance on the Atari benchmark.

---

### Official Review · AnonReviewer3 · 2019-10-26
**Official Blind Review #3**

**Rating:** 6

**Review:**

The paper introduces a new algorithm for parallelizing monte carlo tree search (MCTS). MCTS is hard to parallelize as we have to keep track of the statistics of the node of the tree, which are typically not up-to-date in a parallel execution. The paper introduces a new algorithm that updates the visitation counts before evaluating the rollout (which takes long), and therefore allows other workers to explore different parts of the tree as the exploration bonus is decreased for this node. The algorithm is evaluated on the atari games as well on a proprietary game and compared to other parallelized MCTS variants.

The makes intuitively a lot of sense, albeit it is very simple and it is a surprise that this has not been tried yet. Anyhow, simplicity is not a disadvantage. The algorithm seems to be effective and the evaluations are promising and the paper is also well written. I have only 2 main concerns with the paper:

- The paper is very long (10 pages), and given that, we reviewers should use stricter reviewing rules. As the introduced algorithm is very simple, I do not think that 10 pages are justified. The paper should be considerably shortened (e.g. The "user pass rate prediction system" does not add much to the paper, could be skipped. Moreover, the exact architecture is maybe also not that important).

- The focus of the paper is planning, not learning. Planning conferences such as ICAPS would maybe be a better fit than ICLR.

Given the stricter reviewing guidelines, I am leaning more towards rejects as the algorithmic contribution is small and I do not think 10 pages are justified.

**Experience Assessment:**

I have read many papers in this area.

**Review Assessment: Checking Correctness Of Derivations And Theory:**

I carefully checked the derivations and theory.

**Review Assessment: Checking Correctness Of Experiments:**

I carefully checked the experiments.

**Review Assessment: Thoroughness In Paper Reading:**

I read the paper at least twice and used my best judgement in assessing the paper.

---

> ### Author Response · Authors · 2019-11-10
> **Response to reviewer #3**
>
> Thank you for your valuable comments.
>
> First, we followed your advice to reduce the paper length to 8 pages in the revised version after the following adjustments. (i) We move the experimental results of the “Joy City” that are less relevant to our main point (demonstrating the effectiveness and efficiency of P-UCT) to the supplementary material. This includes descriptions of the “user pass rate prediction system” and relative figures and tables. (ii) We did some minor adjustments such as changing the layout of certain figures to save more space. Together, they make a more compact structure for our 8-page paper.
>
> (note: according to the suggestion by Reviewer #4, we have changed the algorithm name to WU-UCT to avoid confusion with an existing name PUCT, though in the response we still use P-UCT for your convenience.)
>
> Second, we would like to emphasize the main contribution of this paper: it proposes a simple but effective method for parallelizing Monte Carlo Tree Search. As you pointed out, simplicity is not a disadvantage. Although the proposed approach is simple, the idea behind it is non-trivial. As analyzed in Section 4, by keeping track of the unobserved samples, P-UCT manages to avoid common failure modes (e.g. collapse of exploration and exploitation failure, as detailed in Section 4) of other parallel MCTS algorithms. Moreover, with an in-depth empirical analysis with the (unrealistic) ideal parallel algorithm (Figure 1(b)), we show that P-UCT best mimics the sequential algorithm’s behavior compared to other parallelization approaches.
>
> Finally, we think our paper is a good fit for ICLR for the following reasons. First, MCTS is an important component of model-based reinforcement learning and is often combined with learning approaches to achieve better performance (e.g. [1]). Moreover, MCTS has been combined with reinforcement learning methods to learn better policies (e.g. [2]), which indicates that MCTS has been used as a crucial component in learning algorithms. Therefore, though we only evaluated P-UCT under the planning setting, it can be used as part of a learning algorithm. Additionally, as stated by Reviewer #4, “while significant effort has been made by the RL community to scale up distributed model-free algorithms, less effort has been made for model-based algorithms”. P-UCT provides another attempt on scaling up MCTS, an important part of model-based reinforcement learning algorithm, so we think it should be a good fit for ICLR audiences.
>
> [1] Silver, D., Huang, A., Maddison, C. J., Guez, A., Sifre, L., Van Den Driessche, G., ... & Dieleman, S. (2016). Mastering the game of Go with deep neural networks and tree search. nature, 529(7587), 484.
> [2] Silver, D., Schrittwieser, J., Simonyan, K., Antonoglou, I., Huang, A., Guez, A., ... & Chen, Y. (2017). Mastering the game of go without human knowledge. Nature, 550(7676), 354.

---

> > ### Author Response · Authors · 2019-11-15
> > **Additional comments**
> >
> > Besides the above response, we would like to add some further comments regarding the contribution of our work.
> >
> > First, as we pointed out earlier, the simplicity of our algorithm on its implementation side but not on its key idea. Compared to previous works, we address the key challenge of parallelizing MCTS in a more principled manner. Specifically, P-UCT (renamed as WU-UCT) is designed based on the insight that on-going simulations (unobserved samples) will eventually return the results, so their number should be tracked and used to adaptively adjust the UCT selection process. To corroborate that our principled solution performs better than heuristic approaches (e.g., a simple combination of visit count with virtual loss as in [1]), we perform additional experiments to compare with this new baseline (as suggested by Reviewer #4), where the results are copied in the following table. It shows that our principled solution is consistently better than the new baseline, and it does not require task-dependent hyper-parameter tuning.
> >
> > Table. Comparison between WU-UCT and TreeP with both virtual loss (r_vl) and virtual pseudo-count (n_vl). Three sets of hyper-parameters are used in TreeP, and each experiment was repeated two times due to the limited time. (In our final version, we will report the results with 10 runs.) The results of Centipede, Robotank, and NameThisGame are still running and will be reported as soon as they are completed. (These three games are generally more expensive because their game steps are about 10 times of other games.)
> > ======================================================================
> > Env.                      +    WU-UCT             TreeP                      TreeP                      TreeP
> >                              +                          (r_vl = n_vl = 1)      (r_vl = n_vl = 2)      (r_vl = n_vl = 3)
> > ======================================================================
> > Alien                    +    6536±1093        4850±357              4935±60               5000±0
> > Boxing                +        100±0                99±1                       99±0                     99±1
> > Breakout            +       413±14             379±43                  265±50                463±60
> > Freeway              +         32±0                 32±0                      32±0                     32±0
> > Gravitar              +     5060±568         3500±707             4105±463             4950±141
> > MsPacman         +    19804±2232     13160±462          12991±851            8640±438
> > RoadRunner      +   46720±1359      29800±282            28550±459         29400±494
> > Qbert                  +    17953±225       17055±353           13425±194            9075±53
> > SpaceInvaders   +     3000±813        2305±176             3210±127              3020±42
> > Tennis                 +         4±2                    1±0                       1±0                         0±0
> > TimePilot           +   48390±6721      52500±707           49800±212           32400±1697
> > Zaxxon               +   39085±6838     24300±2828          24600±424          37550±1096
> > ======================================================================
> >
> > Second, our work has a high practical value. Despite its outstanding performance, Monte Carlo Tree Search is time-consuming, which brings an urgent need for an effective parallelization algorithm. This work bridge this gap to broaden the application of MCTS, which is confirmed by Section 5.1, where P-UCT (renamed as WU-UCT) is applied successfully in a real-world production system, where ~16 times speedup is achieved with negligible performance loss.
> >
> > Third, we have refined the paper structure as well as including more experiments to make our arguments stronger, and the results further justify the superiority of P-UCT over comparison approaches. The changes during the rebuttal period are summarized below.
> >
> > (i) Based on your suggestion, we have reduced the number of pages to 8 by moving part of the experiment results that are less relevant to our main point (e.g., the “user pass-rate prediction system”) to the supplementary material. We also changed the layout of some figures (e.g., Figure 2 has been redrawn to be more compact as well as clear) to improve the paper’s readability.
> >
> > (ii) We have added comprehensive comparisons with more baseline models in the Atari experiments. Comparisons with the sequential UCT indicates that P-UCT achieves the minimal 16% performance degradation among all parallel algorithms (TreeP: 26%, LeafP: 36%, RootP: 32%) while having ~16 times speedup.
> >
> > (iii) We have adjusted the t-test results with the Bonferroni method (using the p-value threshold 0.05/45 = 0.0011), which is a much stronger requirement. Under this stricter condition, P-UCT is still significantly better than comparison approaches.

---

### Official Review · AnonReviewer4 · 2019-11-03
**Official Blind Review #4**

**Rating:** 8

**Review:**

This paper introduces a new algorithm for parallelizing Monte-Carlo Tree Search (MCTS). Specifically, when expanding a new node in the search tree, the algorithm updates the parent nodes’ statistics of the visit counts but not their values; it is only when the expansion and simulation steps are complete that the values are updated as well. This has the effect of shrinking the UCT exploration term, and making other workers less likely to explore that part of the tree even before the simulation is complete. This algorithm is evaluated in two domains, a mobile game called “Joy City” as well as on Atari. The proposed algorithm results in large speedups compared to serial MCTS with seemingly little impact in performance, and also results in higher scores on Atari than existing parallelization methods.

Scaling up algorithms like MCTS is an important aspect of machine learning research. While significant effort has been made by the RL community to scale up distributed model-free algorithms, less effort has been made for model-based algorithms, so it is exciting to see that emphasis here. Overall I thought the main ideas in paper were clear, the proposed method for how to effectively parallelize MCTS was compelling, and the experimental results were impressive. Thus, I tend to lean towards accept. However, there were three aspects of the paper that I thought could be improved. (1) It was unclear to me how much the parallelization method differs from previous approaches (called “TreeP” in the paper) which adjust both the visit counts and the value estimate. (2) The paper is missing experiments showing the decrease in performance compared to a serial version of the algorithm. (3) The paper did not always provide enough detail and in some cases used confusing terminology. If these three things can be addressed then I would be willing to increase my score.

Note that while I am quite familiar with MCTS, I am less familiar with methods for parallelizing it, though based on a cursory Google Scholar search it seems that the paper is thorough in discussing related approaches.

1. When performing TreeP, does the traversed node also get an increased visit count (in addition to the loss which is added to the value estimate)? In particular, [1] and [2] adjust both the visit counts and the values, which makes them quite similar to the present method (which just adjusts visit counts). It’s not clear from the appendix whether TreeP means that just the values are adjusted, or both the values and nodes. If it is the former, then I would like to see experiments done where TreeP adjusts the visit counts as well, to be more consistent with prior work. (Relatedly, I thought the baselines could be described in significantly more detail than they currently are—-pseudocode would in the appendix would be great!)

2. I appreciate the discussion in Section 4 of how much one would expect the proposed parallelization method to suffer compared to perfect parallelization. However, this argument would be much more convincing if there were experiments to back it up: I want to know empirically how much worse the parallel version of MCTS does in comparison to the serial version of MCTS, controlling for the same number of simulations.

3. While the main ideas in the paper were clear, I thought certain descriptions/terminology were confusing and that some details were missing. Here are some specifics that I would like to see addressed, roughly in order of importance:

- I strongly recommend that the authors choose a different name for their algorithm than P-UCT, which is almost identical (and pronounced the same) as PUCT, which is a frequently used MCTS exploration strategy that incorporates prior knowledge (see e.g. [1] and [2]). P-UCT is also not that descriptive, given that there are other existing algorithms for parallelizing MCTS.

- Generally speaking, it was not clear to me for all the experiments whether they were run for a fixed amount of wallclock time or a fixed number of simulations, and what the fixed values were in either of those cases. The fact that these details were missing made it somewhat more difficult for me to evaluate the experiments. I would appreciate if this could be clarified in the main text for all the experiments.

- The “master-slave” phrasing is a bit jarring due to the association with slavery. I’d recommend using a more inclusive set of terms like “master-worker” or “manager-worker” instead (this shouldn’t be too much to change, since “worker” is actually used in several places throughout the paper already).

- Figure 7c-d: What are game steps? Is this the number of steps taken to pass the level? Why not indicate pass rate instead, which seems to be the main quantity of interest?

- Page 9: are these p-values adjusted for multiple comparisons? If not, please perform this adjustment and update the results in the text. Either way, please also report in the text what adjustment method is used.

- Figure 7: 3D bar charts tend to be hard to interpret (and in some cases can be visually misleading). I’d recommend turning these into heatmaps with a visually uniform colormap instead.

- Page 1, bottom: the first time I read through the paper I did not know what a “user pass-rate” was (until I got to the experiments part of the paper which actually explained this term). I would recommend phrasing this in clearer way, such as “estimating the rate at which users pass levels of the mobile game…”

- One suggestion just to improve the readability of the paper for readers who are not as familiar with MCTS is to reduce the number of technical terms in the first paragraph of the introduction. Readers unfamiliar with MCTS may not know what the expansion/simulation/rollout steps are, or why it’s necessary to keep the correct statistics of the search tree. I would recommend explaining the problem with parallelizing MCTS without using these specific terms, until they are later introduced when MCTS is explained.

- Page 2: states correspond to nodes, so why introduce additional notation (n) to refer to nodes? It would be easier to follow if the same variable (s) was used for both.

Some additional comments:

- Section 5: I’m not sure it’s necessary to explain so much of the detail of the user-pass rate prediction system in the main text. It’s neat that comparing the results of different search budgets of MCTS allow predicting user behavior, but this seems to be a secondary point besides the main point of the paper (which is demonstrating that the proposed parallelization method is effective). I think the right part of Figure 5, as well as Table 1 and Figure 6, could probably go in the supplemental material. As someone with a background in cognitive modeling, I think these results are interesting, but that they are not the main focus of the paper. I was actually confused during my first read through as it was unclear to me initially why the focus had shifted from demonstrating that parallel MCTS works to

- The authors may be interested in [3], which also uses a form of tree search to model human decisions in a game.

- Page 9: the citation to [2] does not seem appropriate here since AlphaGo Zero did not use a pretrained search policy, I think [1] would be correct instead.

[1] Silver, D., Huang, A., Maddison, C. J., Guez, A., Sifre, L., Van Den Driessche, G., ... & Dieleman, S. (2016). Mastering the game of Go with deep neural networks and tree search. nature, 529(7587), 484.
[2] Silver, D., Schrittwieser, J., Simonyan, K., Antonoglou, I., Huang, A., Guez, A., ... & Chen, Y. (2017). Mastering the game of go without human knowledge. Nature, 550(7676), 354.
[3] van Opheusden, B., Bnaya, Z., Galbiati, G., & Ma, W. J. (2016, June). Do people think like computers?. In International conference on computers and games (pp. 212-224). Springer, Cham.

**Experience Assessment:**

I have published one or two papers in this area.

**Review Assessment: Checking Correctness Of Derivations And Theory:**

N/A

**Review Assessment: Checking Correctness Of Experiments:**

I assessed the sensibility of the experiments.

**Review Assessment: Thoroughness In Paper Reading:**

I read the paper at least twice and used my best judgement in assessing the paper.

---

> ### Author Response · Authors · 2019-11-10
> **Response to reviewer #4 (part 1 of 2)**
>
> Thanks a lot for your many constructive feedbacks, which greatly improve our paper.
>
> First, we would like to clarify the difference between our proposed method P-UCT (renamed as WU-UCT based on your suggestion) and Tree Parallelization (TreeP). First of all, all MCTS methods, regardless of being parallel or sequential, would update both the visit counts and the values of the traversed nodes. The key difference is whether they are updated before or after the simulation step is completed. In sequential MCTS, both visit counts and the values are updated AFTER the simulation step is done. In our WU-UCT, the visit counts are updated BEFORE the simulation step completes. To our best knowledge, NONE of the existing TreeP algorithms (or any existing parallel MCTS algorithm) updates the visit counts BEFORE the simulation step finishes. TreeP only updates the values ahead of time using virtual loss. This is also the case for the work [1] and [2]. (Of course, after the simulation step completes, the visit counts in TreeP would be updated in the backpropagation step, just as the sequential MCTS.) For this reason, we do not compare to the variant where both the visit counts and the values are updated ahead of time, since no such variant of TreeP methods exist. As shown in Sections 4-5, our approach (updating counts ahead of time) is better than TreeP (updating values ahead of time by virtual loss) in Sections 4-5. Nevertheless, updating both the values (by virtual loss) and the visit counts BEFORE the simulation step finishes is an interesting case that has not yet been explored. We would like to consider it as a future work. Also, to clarify the algorithm details, we have added the pseudo-codes of our baselines TreeP, LeafP, and RootP in Algorithms 4-6 in Appendix B with detailed descriptions.
>
> Second, we provide additional experiment results for the sequential UCT. The performance of the sequential UCT in the “joy city” game has already been reported in Figure 7, which corresponds to the 1 expansion worker and 1 simulation worker case. For the Atari games, the results of sequential UCT are added as a new column in Table 2. Also, we have added statements in Section 5.2 of the revised paper to show the intention of including the results of the sequential UCT: the performance of sequential UCT is the best we can expect from any parallel UCT algorithm, so we regard it as an upper bound performance of the parallelized algorithms (WU-UCT, TreeP, LeafP, and RootP). For your convenience, we also copy the results of sequential UCT and WU-UCT below. We have completed 12 out of 15 Atari games; the other 3 on-going experiments are more time-consuming (significantly slower than WU-UCT) and we will report them once they are done. On average, WU-UCT has only 16% relative performance loss, which is much smaller than other baselines (TreeP: 26%, LeafP: 36%, RootP: 32%), which supports our analysis in Section 4 that WU-UCT has the closest performance to the sequential UCT.
>
> +=============+==========+===========+===========+=========+
> +  Environment  +      Alien      +      Boxing     +    Breakout   + Freeway   +
> +=============+==========+===========+===========+=========+
> +           UCT         +     6820        +         100        +         462        +        32       +
> +=============+==========+===========+===========+=========+
> +       WU-UCT     +     6538        +         100        +          413       +        32       +
> +=============+==========+===========+===========+=========+
> +=============+==========+===========+===========+=========+
> +  Environment  +    Gravitar   +  MsPacman + RoadRunner +  Qbert      +
> +=============+==========+===========+===========+=========+
> +           UCT         +     4900         +      23021     +         52300    +      17250   +
> +=============+==========+===========+===========+=========+
> +       WU-UCT     +    5060          +      19804     +        46720      +    17953    +
> +=============+==========+===========+===========+=========+
> +=============+=============+=========+==========+=========+
> +  Environment  + SpaceInvaders +    Tennis   +  TimePilot  +     Zaxxon +
> +=============+=============+=========+==========+=========+
> +           UCT         +         3535          +         5        +      52600     +     46800    +
> +=============+=============+=========+==========+=========+
> +       WU-UCT     +         3000          +         4        +      48390     +    39085     +
> +=============+=============+=========+==========+=========+

---

> > ### Author Response · Authors · 2019-11-10
> > **Response to reviewer #4 (part 2 of 2)**
> >
> > Third, thank you for the many comments that improve our paper. We have addressed them carefully one by one, as detailed in the following.
> >
> > (i) Based on your suggestion, we changed our algorithm name from P-UCT to WU-UCT, in order to avoid potential confusion with the existing PUCT algorithm.
> >
> > (ii) All experiments (both in Section 5.1 and 5.2) were run for a fixed number of simulations. Specifically, for the “Joy City” game experiments, a total of 500 simulations were performed, and for the Atari experiments, 128 simulations were performed. We have clarified this in the revised manuscript (in both Sections 5.1 and 5.2).
> >
> > (iii) Based on your suggestion, we have changed the architecture name to “master-worker” in the revised paper.
> >
> > (iv) In Figure 7 (c-d) (which is Figure 4(c-d) in the revised version), game steps refer to the number of steps taken to pass the level and has been clarified in Section 5.1. We have added explanations of the term in the main text. We used game steps instead of pass-rate as the performance indicator because it is a more fine-grained performance metric than the pass-rate. Pass-rate can only indicate whether the agent uses less than a predefined number of steps. For example, if a level is given 20 steps and one agent used on average 10 steps and the other used 15 steps (assume all with low variance), then it will be hard to judge the performance difference between the two agents by using pass-rate alone. In contrast, examining the average game step provides a clear view that the first agent is better.
> >
> > (v) All p-values for t-tests are based on the pairwise comparison (i.e., WU-UCT vs. LeafP, WU-UCT vs. TreeP, and WU-UCT vs. RootP). Therefore, we did not include multiple comparisons, and the p-values are not adjusted for multiple comparisons. We used the term “paired t-test” in Section 5.2 to clarify this.
> >
> > (vi) We have changed the 3D bar charts into heatmaps, and it looks nicer. Thank you!
> >
> > (vii) Based on your suggestion, we have formally defined “user pass-rate” in the revised paper.
> >
> > (viii) Based on your feedback, we have modified the first paragraph of the introduction to make it more accessible for readers less familiar with MCTS.
> >
> > (ix) We have changed the notation of “node” from n to s, which improves the paper’s clarity.
> >
> >
> > Response to the additional comments:
> >
> > (i) Thank you for the suggestion. Initially, we wanted to use the user-pass-rate prediction system as an important motivating application for our WU-UCT algorithm. But we totally agree that the most important experiments in Section 5.1 are the speedup and performance tests across 1, 2, 4, 8, and 16 expansion and simulation workers. Therefore, following your advice, we moved the details about the user-pass rate prediction system and the corresponding performance to the supplementary material.
> >
> > (ii) Thanks again for sharing the interesting work of [3], and we have added discussions on the paper in our revised manuscript. Specifically, it shows how we can capture human behavior and preference using tree search algorithms. By using a board game as a testbed, it captures human preference using a learnable heuristic function and then performs MCTS using the policy specified by the heuristic function. Interestingly, they showed that the MCTS policy well-mimics the human player’s policy and made an important attempt to bridge the gap between human decision-making and computer game playing.
> >
> > (iii) We have corrected the citations of the AlphaGo Zero paper.
> >
> > [1] Silver, D., Huang, A., Maddison, C. J., Guez, A., Sifre, L., Van Den Driessche, G., ... & Dieleman, S. (2016). Mastering the game of Go with deep neural networks and tree search. nature, 529(7587), 484.
> > [2] Silver, D., Schrittwieser, J., Simonyan, K., Antonoglou, I., Huang, A., Guez, A., ... & Chen, Y. (2017). Mastering the game of go without human knowledge. Nature, 550(7676), 354.
> >
> > [3] van Opheusden, B., Bnaya, Z., Galbiati, G., & Ma, W. J. (2016, June). Do people think like computers?. In International conference on computers and games (pp. 212-224). Springer, Cham.

---

> > > ### Comment · AnonReviewer4 · 2019-11-10
> > > **Response to authors**
> > >
> > > Thank you very much for your detailed response, and for the effort you've put into making all these changes. I will have a closer read through the paper again over the next few days (though I already read through the introduction and it sounds great!), but wanted to clarify a few of my comments sooner rather than later:
> > >
> > > 1. On the difference between TreeP and WU-UCT: based on my reading of the AlphaGo paper, it sounds to me like they update both visit counts and value estimates. Specifically, they say:
> > >
> > > "At each in-tree step t ≤ L of the simulation, the rollout statistics are updated as if it had lost −n_vl games, N_r(s_t, a_t) ← N_r(s_t, a_t) + n_vl; W_r(s_t, a_t) ← W_r(s_t , a_t) − n_vl; this virtual loss discourages other threads from simultaneously exploring the identical variation. At the end of the simulation, the rollout statistics are updated in a backward pass through each step t ≤ L, replacing the virtual losses by the outcome, N_r(s_t , a_t) ← N_r(s_t , a_t) − n_vl +1; W_r(s_t , a_t) ← W_r(s_t , a_t) + n_vl + z_t." ("Backup" section in the appendix).
> > >
> > > In Table 5 they say n_vl = 3, so in other words, before performing each simulation they increase the visit counts by 3 and subtract 3 from the number of games that were won. After the simulation is finished, they undo the changes to the visit counts and the value estimate and update them with the actual simulation results. The approach of WU-UCT is still unique, however: I think it is essentially equivalent to updating the visit counts and including a virtual loss which is equal to the current mean estimate (i.e. it is an adaptive virtual loss rather than a fixed virtual loss). But I still think it would be good to clarify this difference, and to compare to this method of implementing the virtual loss.
> > >
> > > (v) Adjustment for multiple comparisons should be performed with all statistical tests, even pairwise tests, in order to control for the family-wise Type I error rate. Specifically, because you are comparing p-values at the threshold of 0.05, if you perform 100 comparisons, then by chance ~5 of those will comparisons will lead you to reject the null hypothesis (i.e. have p values less than 0.05). To handle this issue, it is best practice to adjust the threshold at which you reject the null hypothesis based on the number of comparisons you are performing. A standard way of doing this is the Bonferroni method (https://en.wikipedia.org/wiki/Bonferroni_correction), in which you would divide your target threshold by the number of comparisons you are performing. Based on my understanding of your tests, you compare WU-UCT to the three other methods on 15 Atari games, so you should set your p-value threshold to 0.05/45 = 0.0011.

---

> > > > ### Author Response · Authors · 2019-11-13
> > > > **Response to reviewer #4**
> > > >
> > > > Thank you for your helpful comments.
> > > >
> > > > First, based on your suggestion, we add an additional set of experiments to compare WU-UCT with the new baseline of “TreeP + pre-adjusted visit count + virtual loss” [1] with different hyper-parameters. The experiment results are given in the following table, from which we can see that on 9 out of 12 tasks, WU-UCT outperforms this new baseline (with its best hyper-parameters). Furthermore, we also observe that TreeP does not have an optimal set of hyper-parameters that performs uniformly well on all tasks. In other words, for the “TreeP + pre-adjusted visit count + virtual loss”, the hyper-parameters need to be tuned separately on each individual task. On the other hand, WU-UCT performs consistently well across different tasks. These additional experimental results along with the discussions are also included in Appendix E of the revised paper.
> > > >
> > > > Table. Comparison between WU-UCT and TreeP with both virtual loss (r_vl) and virtual pseudo-count (n_vl). Three sets of hyper-parameters are used in TreeP, and each experiment was repeated two times due to the limited time. (In our final version, we will report the results with 10 runs.) The results of Centipede, Robotank, and NameThisGame are still running and will be reported as soon as they are completed. (These three games are generally more expensive because their game steps are about 10 times of other games.)
> > > > ======================================================================
> > > > Env.                      +    WU-UCT             TreeP                      TreeP                      TreeP
> > > >                              +                          (r_vl = n_vl = 1)      (r_vl = n_vl = 2)      (r_vl = n_vl = 3)
> > > > ======================================================================
> > > > Alien                    +    6536±1093        4850±357              4935±60               5000±0
> > > > Boxing                +        100±0                99±1                       99±0                     99±1
> > > > Breakout            +       413±14             379±43                  265±50                463±60
> > > > Freeway              +         32±0                 32±0                      32±0                     32±0
> > > > Gravitar              +     5060±568         3500±707             4105±463             4950±141
> > > > MsPacman         +    19804±2232     13160±462          12991±851            8640±438
> > > > RoadRunner      +   46720±1359      29800±282            28550±459         29400±494
> > > > Qbert                  +    17953±225       17055±353           13425±194            9075±53
> > > > SpaceInvaders   +     3000±813        2305±176             3210±127              3020±42
> > > > Tennis                 +         4±2                    1±0                       1±0                         0±0
> > > > TimePilot           +   48390±6721      52500±707           49800±212           32400±1697
> > > > Zaxxon               +   39085±6838     24300±2828          24600±424          37550±1096
> > > > ======================================================================
> > > >
> > > > We agree with the reviewer that our proposed WU-UCT is a more principled parallel UCT algorithm when compared with the above TreeP variant (count + virtual loss). Conceptually, WU-UCT is designed based on the fact that on-going simulations (unobserved samples) will eventually return the results, so their number should be tracked and used to adaptively adjust the UCT selection process. On the other hand, TreeP uses an artificially designed virtual loss r_vl and a hand-crafted count correction n_vl to discourage other threads from simultaneously exploring the same node. Therefore, WU-UCT achieves a better exploration-exploitation tradeoff in parallelization, which leads to better performance as confirmed by the above experimental results.
> > > >
> > > > Second, following your suggestion, we have adjusted the t-test results by using the p-value threshold 0.05/45 = 0.0011 and have updated the results in the revised paper. The new results indicate that WU-UCT performs significantly better than TreeP, LeafP, and RootP in 4, 5, and 6 games, respectively. However, note that most experiments are only repeated 3 times at this moment, making it extremely hard to reject the null hypothesis under the threshold 0.0011. We are running additional experiments to repeat all experiments in Table 2 ten times (suggested by Reviewer #2), and will update the t-test results again after that. Finally, we would like to point out that the Bonferroni adjustment method is very conservative as it performs a family hypotheses test, whose p-value is corrected according to the probability of rejecting at least one hypothesis. Nevertheless, even under this much stronger requirement, WU-UCT is still significantly better.
> > > >
> > > > [1] Silver, D., Huang, A., Maddison, C. J., Guez, A., Sifre, L., Van Den Driessche, G., ... & Dieleman, S. (2016). Mastering the game of Go with deep neural networks and tree search. Nature, 529(7587), 484.

---

> > > > > ### Comment · AnonReviewer4 · 2019-11-14
> > > > > **Response to authors**
> > > > >
> > > > > Thank you for including these additional results! I am very glad to see that WU-UCT still outperforms the TreeP baseline even when it adjusts the visit counts. I think this comparison makes the results even more compelling. I am also glad to see the results roughly hold for the adjusted p-values (I agree Bonferroni is a bit conservative, but I feel it is better to err on the side of conservatism for these types of comparisons, just to be sure).
> > > > >
> > > > > I think this paper is a valuable contribution to be presented at ICLR, and I think that all the changes and additional experiments have changed this from a good paper to a great paper. I will thus be increasing my score.

---

### Author Response · Authors · 2019-11-10
**General response to all reviewers**

We thank all the reviewers for their useful feedback. As suggested by Reviewers #3 and #4, we move the paragraphs that are related to the “user pass-rate prediction system” (in Section 5.1) to the supplementary material. In addition, we also did minor adjustments to the layout of some figures. Together, we reduce the paper to 8 pages, as suggested by Reviewer #3, which provides a more compact structure for the paper. For the detailed responses to your comments, please refer to our reply posted under each review comment.

Furthermore, as suggested by Reviewer #4, we also changed our algorithm name from P-UCT to WU-UCT, to differentiate it from the existing PUCT algorithm in [1] and [2].

[1] Silver, D., Huang, A., Maddison, C. J., Guez, A., Sifre, L., Van Den Driessche, G., ... & Dieleman, S. (2016). Mastering the game of Go with deep neural networks and tree search. nature, 529(7587), 484.
[2] Silver, D., Schrittwieser, J., Simonyan, K., Antonoglou, I., Huang, A., Guez, A., ... & Chen, Y. (2017). Mastering the game of go without human knowledge. Nature, 550(7676), 354.

---

### Decision · Program_Chairs · 2019-12-19

**Decision:**

Accept (Talk)

**Comment:**

The paper investigates parallelizing MCTS.
The authors propose a simple method based on only updating the exploration bonus
in (P)-UCT by taking into account the number of currently ongoing / unfinished
simulations.
The approach is extensively tested on a variety of environments, notably
including ATARI games.

This is a good paper.
The approach is simple, well motivated and effective.
The experimental results are convincing and the authors made a great effort to
further improve the paper during the rebuttal period.
I recommend an oral presentation of this work, as MCTS has become a
core method in RL and planning, and therefore I expect a lot of interest in the
community for this work.